# TurboQuant: Online Vector Quantization with Near-optimal Distortion Rate

**Amir Zandieh**
Google Research
zandieh@google.com

Majid Daliri
New York University
daliri.majid@nyu.edu

Majid Hadian
Google DeepMind
majidh@google.com

Vahab Mirrokni
Google Research
mirrokni@google.com

## ABSTRACT

Vector quantization, a problem rooted in Shannon's source coding, aims to quantize high-dimensional Euclidean vectors while minimizing distortion in their geometric structure. We propose TurboQuant to address both mean-squared error (MSE) and inner product distortion, overcoming limitations of existing methods that fail to achieve optimal distortion rates. Our data-oblivious algorithms, suitable for online applications, achieve near-optimal distortion rates (within a small constant factor) across all bit-widths and dimensions. TurboQuant achieves this by randomly rotating input vectors, inducing a concentrated Beta distribution on coordinates, and leveraging the near-independence of distinct coordinates in high dimensions to simply apply optimal scalar quantizers per each coordinate. Recognizing that MSE-optimal quantizers introduce bias in inner product estimation, we propose a two-stage approach: applying an MSE quantizer followed by a 1-bit Quantized JL (QJL) transform on the residual, resulting in an unbiased inner product quantizer. We also provide a formal proof of the information-theoretic lower bounds on best achievable distortion rate by any vector quantizer, demonstrating that TurboQuant closely matches these bounds, differing only by a small constant ($\approx 2.7$) factor. Experimental results validate our theoretical findings, showing that for KV cache quantization, we achieve absolute quality neutrality with 3.5 bits per channel and marginal quality degradation with 2.5 bits per channel. Furthermore, in nearest neighbor search tasks, our method outperforms existing product quantization techniques in recall while reducing indexing time to virtually zero.

Vector quantization (VQ) in Euclidean space is crucial for efficiently handling high-dimensional vectors across a spectrum of computational domains, from training and deploying large-scale AI and deep learning models to powering vector databases for search/retrieval. The core objective is to compress high-dimensional vectors by quantizing them–converting floating-point coordinate values to low-bitwidth integers–while minimizing distortion, quantified by metrics such as mean-squared error (MSE) or inner product errors. By preserving these properties, inner product queries can be answered rapidly, with minimal latency, and using reduced computational and communication resources.

This problem traces back to Shannon's seminal work on source coding Shannon (1948); Shannon et al. (1959), which established that the minimum distortion achievable by block source codes—now known as vector quantizers—is characterized by the distortion–rate function, determined by the source statistics and the chosen distortion measure (e.g., MSE). Today, vector quantization plays a central role in modern computational domains, including AI, deep learning, and large-scale search systems.

A key application of VQ is in deployment of AI models, including LLMs Achiam et al. (2023); Dubey et al. (2024); Anthropic (2024); Team et al. (2024). As LLM capabilities depend heavily on their model size and context length Kaplan et al. (2020), serving them requires substantial memory demands and increased inference latency. This latency is primarily attributed to communication bot-

tlenecks between HBM and SRAM on accelerators, or across distributed clusters. By compressing or quantizing model weights and activations, we can effectively mitigate these bottlenecks, resulting in significant reductions in inference costs. Inner product operations between activations and weights is at the core of deep learning models. Thus, model quantization schemes strive to compress weights and/or activation vectors while accurately preserving these inner products.

Decoder based transformer models Vaswani et al. (2017) present another compelling use case. These models must store key/value (KV) embeddings from previously generated tokens in the KV cache, the size of which scales with both model size (number of layers and attention heads) and context length. This scaling is a significant bottleneck in terms of memory usage and computational speed, especially for long context models. Therefore, reducing the KV cache size without compromising accuracy is essential. In this context, the preservation of the Euclidean structure of these embedding vectors–their inner products and distances–is crucial for maintaining model performance. VQ emerges as the most suitable framework for addressing this challenge, offering a robust approach to compressing high-dimensional embeddings while preserving their essential geometric properties.

Additionally, nearest neighbor (NN) search in high-dimensional spaces with inner product or cosine similarity ela (2025); Guo et al. (2020) is a cornerstone of vector databases pin (2025); gdr (2025); pgv (2025). These databases are fundamental for retrieval-augmented generation Gao et al. (2023); Edge et al. (2024) and information retrieval Khattab & Zaharia (2020); Santhanam et al. (2021). VQ, a.k.a. product quantization (PQ), plays a critical role by enabling efficient compression of database vectors, optimizes memory usage, and facilitates low-latency, accurate estimations of inner products with query vectors, thereby enabling fast and precise nearest neighbor searches.

Existing VQ algorithms present a trade-off: either they lack accelerator (vectorization) compatibility and exhibit slow computation, making them unsuitable for real-time AI applications like KV cache quantization, or they lack distortion guarantees as sharp as ours relative to bit-width. Our objective is to introduce an algorithm that addresses these limitations. Specifically, we design TURBOQUANT: a lightweight, capable of online application (crucial for scenarios like KV cache quantization), and highly accelerator-friendly—a critical attribute for modern AI workloads.

The core of TURBOQUANT is a two-stage process. First, we develop a quantizer with optimal distortion rate in terms of mean-squared error (MSE). Subsequently, we apply a 1-bit quantizer to the residual, resulting in an unbiased and low-distortion inner product quantizer. We demonstrate that quantizers optimized for MSE do not produce unbiased estimators for inner products, and our two-stage solution effectively bridges this gap. Our quantizer starts by randomly rotating $d$-dimensional input vectors. Random rotation has been an effective technique in quantization and low-precision transformer literature, and has been used multiple times previously, e.g., in Chee et al. (2024); Xi et al. (2023); Ashkboos et al. (2024b). Observing the key fact that each coordinate in the rotated vectors follows a Beta distribution, we design optimal Lloyd-Max quantizer Lloyd (1982); Max (1960) for each coordinate by solving a continuous k-means problem. This method gives optimal MSE distortion bound and minimizes the L2 norm of the residual. To obtain an unbiased and low-distortion quantizer for inner products, we compose our quantizer with the recently developed Quantized Johnson-Lindenstrauss (QJL) transform Zandieh et al. (2024a), which quantizes each coordinate of the residual vector to a single bit. Our algorithm offers provably optimal distortion bounds for both MSE and inner products, achieving an exponential improvement over existing methods in terms of bit-width dependence.

## 0.1 PROBLEM DEFINITION

Formally, our goal is to design a quantization map, denoted as $Q : \mathbb{R}^d \to \{0, 1\}^B$, that transforms $d$-dimensional vectors to $B$-bit binary strings. If we set $B = b \cdot d$ for some $b \geq 0$, this quantizer is said to have a bit-width of $b$, representing the average number of bits used to encode each coordinate of $\mathbb{R}^d$. Crucially, we require an inverse map, $Q^{-1} : \{0, 1\}^B \to \mathbb{R}^d$ that performs dequantization, approximately reconstructing original vectors from their quantized representations. Of course, this transformation is inherently lossy, as $Q$ is not a bijection. So, our primary objective is to minimize distortion, with a specific focus on mean-squared error (MSE) and inner product distortion.

We make no distributional assumptions on the input vectors, treating them in the worst-case setting. We let the quantizer $Q(\cdot)$ to be randomized, leading to stochastic outputs. Considering randomized quantizers, it is more appropriate to define the expected distortion over the randomness of the

quantizer's output. Thus, we aim to design quantizers that for any desired bit-width $b$ minimize the following expected distortion measures for any (worst-case) vectors $\boldsymbol{x}, \boldsymbol{y} \in \mathbb{R}^d$:

$$\textbf{(MSE)} \quad D_{\texttt{mse}} := \underset{Q}{\mathbb{E}} \left[ \left\| \boldsymbol{x} - Q^{-1}\left(Q(\boldsymbol{x})\right) \right\|_2^2 \right] \tag{1}$$

$$\textbf{(inner-prod error)} \quad D_{\texttt{prod}} := \underset{Q}{\mathbb{E}} \left[ \left| \langle \boldsymbol{y}, \boldsymbol{x} \rangle - \langle \boldsymbol{y}, Q^{-1}\left(Q(\boldsymbol{x})\right) \rangle \right|^2 \right]. \tag{2}$$

The expectations above are taken with respect to the randomness of the quantizer $Q(\cdot)$. Furthermore, for inner-product quantizers, we require unbiasedness of the inner product estimator, a desirable property for numerous applications. More precisely, we require:

$$\textbf{(unbiased inner-prod)} \quad \underset{Q}{\mathbb{E}} \left[ \langle \boldsymbol{y}, Q^{-1}\left(Q(\boldsymbol{x})\right) \rangle \right] = \langle \boldsymbol{y}, \boldsymbol{x} \rangle.$$

We aim to design computationally efficient quantizers $Q_{\texttt{mse}}$ and $Q_{\texttt{prod}}$, that achieve optimal bounds for the distortion measures defined above, for any given bit-width $b$. Additionally, we aim for $Q_{\texttt{prod}}$ to provide unbiased inner product estimates. In particular, assume that we are given $n$ real-valued vectors $\boldsymbol{x}_1, \boldsymbol{x}_2, \dots \boldsymbol{x}_n \in \mathbb{R}^d$. We design the following primitives:

- QUANT: efficiently quantizes the dataset and computes $Q(\boldsymbol{x}_1), Q(\boldsymbol{x}_2), \dots Q(\boldsymbol{x}_n)$.
- DEQUANT: efficiently reconstructs original vectors by computing $Q^{-1}\left(Q(\boldsymbol{x}_i)\right)$ for $i \in [n]$.

## 0.2 PRIOR WORK

Online (data-oblivious) quantization methods apply instantly without needing data-specific tuning or calibrations Dettmers et al. (2022); Ashkboos et al. (2024a); Liu et al. (2024b); Shah et al. (2024); Han et al. (2025a). In contrast, offline (data-dependent) methods require heavy preprocessing and learning to adapt the quantization map to the data, making them unsuitable for dynamic data scenarios Kim et al. (2023); Nikdan et al. (2026). For instance, methods such as those presented in Frantar et al. (2022); Lin et al. (2024); Xiao et al. (2023a); Chee et al. (2023) use second-order (Hessian) information to tune the quantization map which requires heavy preprocessing and even in some cases post processing as well.

Due to space constraints, we include a more detailed discussion of prior work in Appendix A.

## 0.3 OVERVIEW OF TECHNIQUES AND CONTRIBUTIONS

**MSE Optimized TURBOQUANT.** Our first VQ algorithm is designed to minimize MSE distortion defined in Eq. (1). To achieve this, we apply a random rotation to the input vectors, thereby inducing a Beta distribution on each coordinate, irrespective of the input vectors themselves. In high dimensions $d$, the distribution of each coordinate converges to a Normal distribution due to concentration of measure and the central limit theorem. Furthermore, any two distinct coordinates become nearly uncorrelated and, more importantly, almost independent (a deeper result that goes beyond just correlation). This near-independence is a crucial aspect that simplifies our quantization design. It allows us to quantize each coordinate using optimal scalar quantization, disregarding interactions or correlations between different coordinates, while still achieving near-optimal distortion.

We find optimal scalar quantizers for random variables with Beta distributions by solving a continuous 1-d k-means problem using the Max-Lloyd algorithm. We precompute and store these optimal codebooks for a range of practically useful bit-widths, to enable efficient subsequent invocations of our TURBOQUANT algorithm. In Theorem 1 we prove that the $b$-bit MSE optimized TURBO-QUANT $Q_{\texttt{mse}} : \mathbb{R}^d \to \{0,1\}^{b \cdot d}$ achieves the following distortion for any worst-case vector $\boldsymbol{x} \in \mathbb{R}^d$ with $\|\boldsymbol{x}\| = 1$, without any assumption on the distribution of $\boldsymbol{x}$:

- $D_{\texttt{mse}}(Q_{\texttt{mse}}) := \mathbb{E}\left[ \left\| \boldsymbol{x} - Q_{\texttt{mse}}^{-1}\left(Q_{\texttt{mse}}(\boldsymbol{x})\right) \right\|_2^2 \right] \le \frac{\sqrt{3}\pi}{2} \cdot \frac{1}{4^b}$ for any $b \ge 0$.
- For small bit-widths the above distortion upper bound can be further refined. Specifically, for $b = 1, 2, 3, 4$ we have $D_{\texttt{mse}}(Q_{\texttt{mse}}) \approx \mathbf{0.36}, \mathbf{0.117}, \mathbf{0.03}, \mathbf{0.009}$, respectively.

Note that the unit norm assumption, $\|x\|_2 = 1$, is standard and not restrictive and can compute and store the $L2$ norms in floating-point precision and rescale the dequantized points.

**Inner Product TURBOQUANT.** We show that the MSE optimized quantizers are biased for inner product estimation, and thus a different VQ scheme is needed to get an unbiased inner product quantizer. Our solution is a two-stage algorithm that first applies the above-mentioned $Q_{\text{mse}}$ with a bit-width one less than our target budget and then applies a QJL Zandieh et al. (2024a) on the residual error. This is proven to be unbiased and also has a nearly optimal inner product error rate. In Theorem 2 we prove that the $b$-bit inner product optimized TURBOQUANT $Q_{\text{prod}} : \mathbb{R}^d \rightarrow \{0,1\}^{b \cdot d}$ achieves the following distortion for any vectors $\boldsymbol{x}, \boldsymbol{y} \in \mathbb{R}^d$ with $\|\boldsymbol{x}\| = 1$, without any assumption on the distribution of $\boldsymbol{x}, \boldsymbol{y}$:

- $\mathbb{E}\left[\langle \boldsymbol{y}, Q_{\text{prod}}^{-1}\left(Q_{\text{prod}}(\boldsymbol{x})\right)\rangle\right] = \langle \boldsymbol{y}, \boldsymbol{x} \rangle$

- $D_{\text{prod}}(Q_{\text{prod}}) := \mathbb{E}\left[\left|\langle \boldsymbol{y}, \boldsymbol{x} \rangle - \langle \boldsymbol{y}, Q_{\text{prod}}^{-1}\left(Q_{\text{prod}}(\boldsymbol{x})\right)\rangle\right|^2\right] \leq \frac{\sqrt{3}\pi^2 \cdot \|\boldsymbol{y}\|_2^2}{d} \cdot \frac{1}{4^b}$ for any $b \geq 0$.

- For small bit-widths the above distortion upper bound can be further refined. Specifically, for $b = 1, 2, 3, 4$ we have $D_{\text{prod}}(Q_{\text{prod}}) \approx \frac{1.57}{d}, \frac{0.56}{d}, \frac{0.18}{d}, \frac{0.047}{d}$, respectively.

**Lower Bound.** As shown in Theorem 3 in the appendix, we leverage Shannon's lower bound and Yao's minimax principle to prove that for any randomized quantization algorithm $Q : \mathbb{R}^d \rightarrow \{0,1\}^{b \cdot d}$ with bit-width $b$, there exist hard input instances $\boldsymbol{x}, \boldsymbol{y} \in \mathbb{R}^d$ with $\|\boldsymbol{x}\| = 1$ such that the following lower bounds hold:

- $D_{\text{mse}}(Q) := \mathbb{E}\left[\left\|\boldsymbol{x} - Q^{-1}\left(Q(\boldsymbol{x})\right)\right\|_2^2\right] \geq \frac{1}{4^b}$

- $D_{\text{prod}}(Q) = \mathbb{E}\left[\left|\langle \boldsymbol{y}, \boldsymbol{x} \rangle - \langle \boldsymbol{y}, Q^{-1}\left(Q(\boldsymbol{x})\right)\rangle\right|^2\right] \geq \frac{\|\boldsymbol{y}\|_2^2}{d} \cdot \frac{1}{4^b}$

As demonstrated by our lower bounds, TURBOQUANT's MSE distortion is provably within a factor of at most $\frac{\sqrt{3}\pi}{2} \approx \boldsymbol{2.7}$ of the information-theoretical lower bound. Notably, for smaller bit-widths, this factor significantly decreases. For instance, at a bit-width of $b = 1$ TURBOQUANT achieves a distortion that is only a factor of approximately $\boldsymbol{1.45}$ away from the optimal which is also confirmed by our experimental results, indicating its efficiency in low-bit-width scenarios.

**Experimental Results.** In Section 2.1, we empirically validate our theoretical distortion bounds, demonstrating that TURBOQUANT's observed distortions closely align with our predictions across various real-world datasets, approaching the established lower bounds. Furthermore, in Section 2.2 and Section 2.3, we showcase TURBOQUANT's efficacy in online KV cache quantization. Specifically, we achieve perfect long-context retrieval in needle-in-a-haystack tasks as well as other long-context tasks, while compressing the KV cache by a factor exceeding $5\times$. Finally in Section 2.4 we apply TURBOQUANT to various high-dimensional near neighbor search tasks. TURBOQUANT consistently outperforms data-dependent product quantization, while reducing the indexing time to near zero.

**Notations.** We use boldface lowercase letters, such as $\boldsymbol{x}$ and $\boldsymbol{y}$, to denote vectors, and boldface uppercase letters, like $\boldsymbol{M}$, to denote matrices. To denote a slice of a vector $\boldsymbol{x}$ between the coordinate indices $i$ and $j$ inclusive of the endpoints, we use $\boldsymbol{x}_{i:j}$. For a matrix $\boldsymbol{M}$, we simply write $\boldsymbol{M}_i$ to denote its $i$-th row vector. We use the notation $\mathbb{S}^{d-1}$ to denote the hypersphere in $\mathbb{R}^d$ of radius 1.

# 1 TURBOQUANT: HIGH PERFORMANCE QUANTIZATION

We developed two VQ algorithms, each tailored to a specific objective. The first algorithm is designed to minimize the MSE between the original and reconstructed vectors after quantization. The second algorithm is optimized for unbiased inner product estimation, addressing the bias inherent in MSE-optimal quantizers. These algorithms are detailed in the following subsections.

Furthermore, in Appendix C.3, we establish information-theoretic lower bounds on the best achievable distortion rates for any vector quantizer. This analysis demonstrates that TURBOQUANT achieve near-optimality, differing from the lower bound by only a small constant factor across all bit-widths.

---

**Algorithm 1** TURBOQUANT$_{\texttt{mse}}$: optimized for MSE

---

1: **input:** dimension $d$ and bit-width $b$
2: Generate a `random rotation matrix` $\mathbf{\Pi} \in \mathbb{R}^{d \times d}$
3: Construct `codebook` by finding centroids $c_1, c_2, \dots c_{2^b} \in [-1, 1]$ that minimize MSE cost in Eq. (3)

---

4: **Procedure** QUANT$_{\texttt{mse}}(\boldsymbol{x})$
5: $\boldsymbol{y} \leftarrow \mathbf{\Pi} \cdot \boldsymbol{x}$
6: $\texttt{idx}_j \leftarrow \arg\min_{k \in [2^b]} |\boldsymbol{y}_j - c_k|$ for every $j \in [d]$      {`idx`$_j$'s are $b$-bit integers}
7: **output:** idx

---

8: **Procedure** DEQUANT$_{\texttt{mse}}(\texttt{idx})$
9: $\tilde{\boldsymbol{y}}_j \leftarrow c_{\texttt{idx}_j}$ for every $j \in [d]$
10: $\tilde{\boldsymbol{x}} \leftarrow \mathbf{\Pi}^\top \cdot \tilde{\boldsymbol{y}}$
11: **output:** $\tilde{\boldsymbol{x}}$

---

## 1.1 MSE OPTIMAL TURBOQUANT

Let $\boldsymbol{x} \in \mathbb{S}^{d-1}$ be a (worst-case) vector on the unit sphere in dimension $d$. We aim to quantize $\boldsymbol{x}$ to $b$ bits per coordinate while minimizing the reconstruction MSE defined in Eq. (1). We start by randomizing this vector by multiplying it with a random rotation matrix $\mathbf{\Pi} \in \mathbb{R}^{d \times d}$. We can generate $\mathbf{\Pi}$ by applying QR decomposition on a random matrix with i.i.d Normal entries.

The resulting rotated vector, $\mathbf{\Pi} \cdot \boldsymbol{x}$, is uniformly distributed on the unit sphere $\mathbb{S}^{d-1}$. As shown in Lemma 1, each coordinate of $\mathbf{\Pi} \cdot \boldsymbol{x}$ follows a Beta distribution, which converges to a normal distribution in high dimensions. Furthermore, in high dimensions, distinct coordinates of $\mathbf{\Pi} \cdot \boldsymbol{x}$ become nearly independent Vershynin (2018), allowing us to apply optimal scalar quantizers to each coordinate independently. Therefore, by Lemma 1, our task reduces to designing a scalar quantizer for random variables with the distribution $f_X(x) = \frac{\Gamma(d/2)}{\sqrt{\pi} \cdot \Gamma((d-1)/2)} \left(1 - x^2\right)^{(d-3)/2}$ for $x \in [-1, 1]$.

The optimal scalar quantization problem, given a known probability distribution, can be framed as a continuous k-means problem in dimension one. Specifically, we aim to partition the interval $[-1, 1]$ into $2^b$ clusters/buckets. The optimal solution adheres to a Voronoi tessellation Lloyd (1982), meaning interval boundaries are the midpoints between consecutive centroids, when arranged in sorted order. Therefore, with $c_i$'s denoting the centroids in ascending order, we can formulate the scalar quantization as the following k-means optimization problem:

$$\mathcal{C}(f_X, b) := \min_{-1 \le c_1 \le c_2 \le \dots \le c_{2^b} \le 1} \sum_{i=1}^{2^b} \int_{\frac{c_{i-1} + c_i}{2}}^{\frac{c_i + c_{i+1}}{2}} |x - c_i|^2 \cdot f_X(x) \, dx. \quad (3)$$

Note that $\mathcal{C}(f_X, b)$ in Eq. (3) denotes the optimal MSE cost function for bit-width $b$, a quantity we will bound to prove the upper bound on the end-to-end MSE of TURBOQUANT. The problem in Eq. (3) can be solved using iterative numerical methods to achieve any desired precision. We solve Eq. (3) for a range of practically relevant bit-widths $b$ once, and store the results for future uses by the quantizer. For example, in moderately high dimensions $d$, where the distribution $f_X(x)$ closely approximates a normal distribution, the optimal quantization centroids for bit-widths $b = 1, 2$ are $\left\{ \pm \frac{\sqrt{2/\pi}}{\sqrt{d}} \right\}$ and $\left\{ \pm \frac{0.453}{\sqrt{d}}, \pm \frac{1.51}{\sqrt{d}} \right\}$, respectively.

Therefore the quantizer $Q_{\texttt{mse}} : \mathbb{R}^d \to \{0, 1\}^{b \cdot d}$ first computes $\mathbf{\Pi} \cdot \boldsymbol{x}$ and then computes and stores the indices of the nearest centroids to each coordinate of this vector. The dequantization map $Q_{\texttt{mse}}^{-1} : \{0, 1\}^{b \cdot d} \to \mathbb{R}^d$ reconstructs the vector by retrieving the centroids corresponding to the stored indices and then rotating the result back to the original basis through multiplication with $\mathbf{\Pi}^\top$. A pseudocode for these procedures is given in Algorithm 1.

We are now ready to prove our main theorem for TURBOQUANT$_{\texttt{mse}}$, the proof is in Appendix C.

**Theorem 1** (Performance Guarantee: TURBOQUANT$_{\texttt{mse}}$)**.** *For any bit-width $b \ge 1$ and any vector $\boldsymbol{x} \in \mathbb{S}^{d-1}$, the procedure* QUANT$_{\texttt{mse}}(\boldsymbol{x})$ *in Algorithm 1 outputs an index vector* $\texttt{idx} \in [2^b]^d$*. When*

---

**Algorithm 2** TURBOQUANT$_{\text{prod}}$: optimized for inner product

---

1: **input:** dimension $d$ and bit-width $b$
2: Instantiate a `TURBOQUANT`$_{\text{mse}}$ with bit-width $b-1$ as per Algorithm 1
3: Generate a `random projection matrix` $\boldsymbol{S} \in \mathbb{R}^{d \times d}$ with i.i.d. entries $\boldsymbol{S}_{i,j} \sim \mathcal{N}(0,1)$

---

4: **Procedure** QUANT$_{\text{prod}}(\boldsymbol{x})$
5: $\texttt{idx} \leftarrow$ QUANT$_{\text{mse}}(\boldsymbol{x})$
6: $\boldsymbol{r} \leftarrow \boldsymbol{x} -$ DEQUANT$_{\text{mse}}(\texttt{idx})$ $\qquad\qquad\qquad\qquad\qquad$ {residual vector}
7: $\texttt{qjl} \leftarrow \text{sign}\,(\boldsymbol{S} \cdot \boldsymbol{r})$ $\qquad\qquad\qquad\qquad\qquad\qquad$ {QJL on residual vector}
8: **output:** $(\texttt{idx}, \texttt{qjl}, \|\boldsymbol{r}\|_2)$

---

9: **Procedure** DEQUANT$_{\text{prod}}(\texttt{idx}, \texttt{qjl}, \gamma)$
10: $\tilde{\boldsymbol{x}}_{\text{mse}} \leftarrow$ DEQUANT$_{\text{mse}}(\texttt{idx})$
11: $\tilde{\boldsymbol{x}}_{\text{qjl}} \leftarrow \frac{\sqrt{\pi/2}}{d} \cdot \gamma \cdot \boldsymbol{S}^\top \cdot \texttt{qjl}$
12: **output:** $\tilde{\boldsymbol{x}}_{\text{mse}} + \tilde{\boldsymbol{x}}_{\text{qjl}}$

---

*this index vector is passed to the primitive* DEQUANT$_{\text{mse}}(\texttt{idx})$, *it produces a reconstructed vector* $\tilde{\boldsymbol{x}} \in \mathbb{R}^d$ *that satisfies the following distortion bounds:*

- *MSE defined as $D_{\text{mse}} := \mathbb{E}_{\tilde{\boldsymbol{x}}}[\|\boldsymbol{x} - \tilde{\boldsymbol{x}}\|_2^2]$ is bounded by $D_{\text{mse}} \leq \frac{\sqrt{3}\pi}{2} \cdot \frac{1}{4^b}$ for any $b \geq 0$.*

- *For small bit-widths, specifically $b = 1, 2, 3, 4$ the MSE exhibits finer-grained distortion values: $D_{\text{mse}} \approx \mathbf{0.36}, \mathbf{0.117}, \mathbf{0.03}, \mathbf{0.009}$, respectively.*

## 1.2 INNER-PRODUCT OPTIMAL TURBOQUANT

For important applications like nearest neighbor search, having an unbiased inner product estimator is essential. However, TURBOQUANT$_{\text{mse}}$ presented in Section 1.1 does not provide unbiased inner product estimates with query vectors. To illustrate this, consider the case with a bit-width of $b = 1$. In this scenario, the optimal codebooks that solve the optimization problem in Eq. (3), for sufficiently large $d$, are $\left\{\pm\sqrt{2/\pi d}\right\}$. This implies that the quantization map for TURBOQUANT$_{\text{mse}}$ is $Q_{\text{mse}}(\boldsymbol{x}) = \text{sign}\,(\boldsymbol{\Pi} \cdot \boldsymbol{x})$ for any $\boldsymbol{x} \in \mathbb{R}^d$, and the dequantization map is $Q_{\text{mse}}^{-1}(\boldsymbol{z}) = \sqrt{2/\pi d} \cdot \boldsymbol{\Pi}^\top \cdot \boldsymbol{z}$ for any $\boldsymbol{z} \in \{-1, +1\}^d$. Therefore, for large enough $d$, according to Lemma 4, we have $\mathbb{E}\left[\langle \boldsymbol{y}, Q_{\text{mse}}^{-1}(Q_{\text{mse}}(\boldsymbol{x}))\rangle\right] = \frac{2}{\pi} \cdot \langle \boldsymbol{y}, \boldsymbol{x}\rangle$, which has a multiplicative bias of $2/\pi$. This bias diminishes with increasing bit-widths $b$, as we empirically demonstrate in Section 2.1.

To address this bias, we propose a solution that combines TURBOQUANT$_{\text{mse}}$ with an instance of QJL Zandieh et al. (2024a). Specifically, let $Q_{\text{mse}}$ be the quantization map corresponding to TURBOQUANT$_{\text{mse}}$ with a bit-width of $b - 1$. For any $\boldsymbol{x} \in \mathbb{S}^{d-1}$ the residual vector, defined as $\boldsymbol{r} := \boldsymbol{x} - Q_{\text{mse}}^{-1}(Q_{\text{mse}}(\boldsymbol{x}))$, has a small L2 norm, i.e., on expectation $\mathbb{E}[\|\boldsymbol{r}\|] = \sqrt{\mathcal{C}(f_X, b-1)}$ (per Eq. (3)). We can then apply the QJL quantization map $Q_{\text{qjl}}$ on this residual vector, resulting in an overall bit-width of $b$ and providing the following unbiased inner product estimator:

$$\langle \boldsymbol{y}, Q_{\text{mse}}^{-1}(Q_{\text{mse}}(\boldsymbol{x}))\rangle + \|\boldsymbol{r}\|_2 \cdot \langle \boldsymbol{y}, Q_{\text{qjl}}^{-1}(Q_{\text{qjl}}(\boldsymbol{r}))\rangle.$$

More formally, the quantization map $Q_{\text{prod}} : \mathbb{S}^{d-1} \to [2^{b-1}]^d \times \{-1, 1\}^d \times \mathbb{R}$ is defined as:

$$Q_{\text{prod}}(\boldsymbol{x}) = \left[Q_{\text{mse}}(\boldsymbol{x}), Q_{\text{qjl}}\left(\boldsymbol{x} - Q_{\text{mse}}^{-1}(Q_{\text{mse}}(\boldsymbol{x}))\right), \left\|\boldsymbol{x} - Q_{\text{mse}}^{-1}(Q_{\text{mse}}(\boldsymbol{x}))\right\|_2\right].$$

A pseudocode for this procedure is given in Algorithm 2.

We prove the main result for TURBOQUANT$_{\text{prod}}$ in the following theorem and prove it in Appendix C.

**Theorem 2** (Performance Guarantee: TURBOQUANT$_{\text{prod}}$)**.** *For any bit-width $b \geq 1$ and any vector $\boldsymbol{x} \in \mathbb{S}^{d-1}$, the procedure QUANT$_{\text{prod}}(\boldsymbol{x})$ in Algorithm 2 outputs an index vector $\texttt{idx} \in [2^{b-1}]^d$ along with a sign vector $\texttt{qjl} \in \{-1, 1\}^d$ and a positive number $\gamma \geq 0$. When these vectors and the scalar value are passed to the primitive DEQUANT$_{\text{prod}}(\texttt{idx}, \texttt{qjl}, \gamma)$, it produces a reconstructed vector $\tilde{\boldsymbol{x}} \in \mathbb{R}^d$ that for any vector $\boldsymbol{y} \in \mathbb{R}^d$ satisfies the following properties:*

- *Expected inner-product $\mathbb{E}_{\tilde{\boldsymbol{x}}}[\langle \boldsymbol{y}, \tilde{\boldsymbol{x}} \rangle] = \langle \boldsymbol{y}, \boldsymbol{x} \rangle$*

- *Inner-product distortion defined as $D_{\mathrm{prod}} := \mathbb{E}_{\tilde{\boldsymbol{x}}}\left[|\langle \boldsymbol{y}, \boldsymbol{x} \rangle - \langle \boldsymbol{y}, \tilde{\boldsymbol{x}} \rangle|^2\right]$ is bounded by $D_{\mathrm{prod}} \leq \frac{\sqrt{3}\pi^2 \cdot \|\boldsymbol{y}\|_2^2}{d} \cdot \frac{1}{4^b}$ for any $b \geq 0$.*

- *For small bit-widths, specifically $b = 1, 2, 3, 4$, $D_{\mathrm{prod}}$ exhibits finer-grained distortion values: $D_{\mathrm{prod}} \approx \frac{1.57}{d}, \frac{0.56}{d}, \frac{0.18}{d}, \frac{0.047}{d}$, respectively.*

## 2 EXPERIMENTS

All experiments are performed using a single NVIDIA A100 GPU. The first set of experiments empirically validates the theoretical results, and the second evaluates the performance of our methods on downstream tasks, specifically KV cache quantization and nearest neighbor vector search.

### 2.1 EMPIRICAL VALIDATION

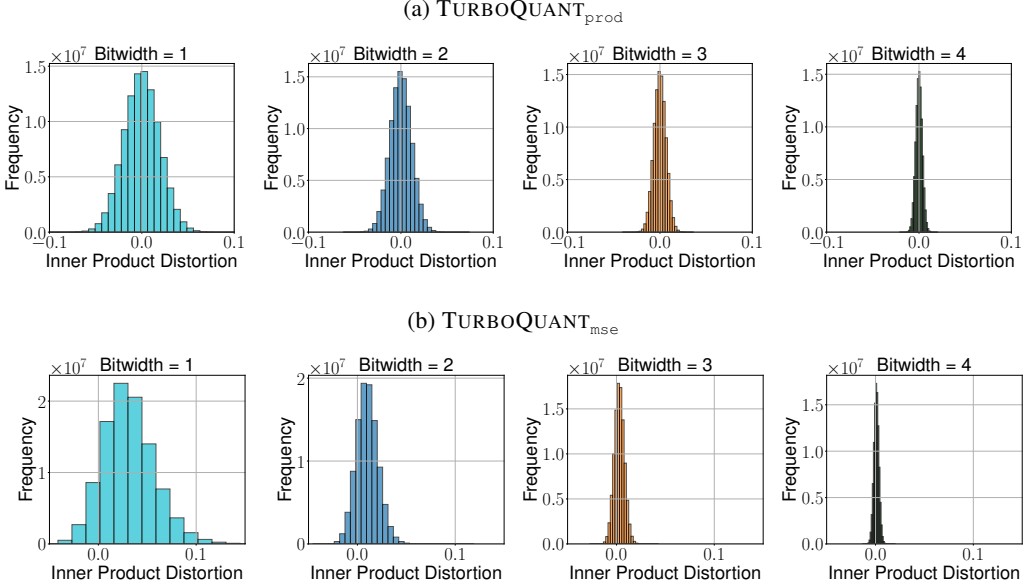

Figure 1: Distribution of Inner Product error for TURBOQUANT$_{\mathrm{prod}}$ and TURBOQUANT$_{\mathrm{mse}}$.

In this section, we validate the theoretical results using experiments on the DBpedia Entities dataset, which is embedded in a 1536-dimensional space via OpenAI3 embeddings. We randomly sample 100,000 points as the training set and extract 1,000 distinct entries as the query set.

We compare two quantization methods: TURBOQUANT$_{\mathrm{prod}}$, optimized for unbiased inner product estimation, and TURBOQUANT$_{\mathrm{mse}}$, which minimizes mean squared error (MSE) between quantized and original vectors. Both methods are applied to estimate inner products by quantizing the training set and analyzing distortion across varying bit widths. As shown in Fig. 1, increasing the bit widths reduces variance in both methods. However, TURBOQUANT$_{\mathrm{mse}}$ introduces bias in inner product estimation, which diminishes and converges to zero with higher bit widths. In contrast, TURBOQUANT$_{\mathrm{prod}}$ remains unbiased across all bit widths, confirming the theoretical guarantees.

In addition to histograms, we also plot in Fig. 2 the average inner product error and MSE between the original and quantized vectors for different bit-widths. These plots are drawn against the upper and lower bounds established in our theoretical analysis and confirm that the results align with the theoretical predictions. Specifically, for inner product estimation, the TURBOQUANTprod approach performs better at lower bit ratios. However, as the bit count increases, TURBOQUANTmse reduces bias and ultimately achieves superior performance in inner product estimation.

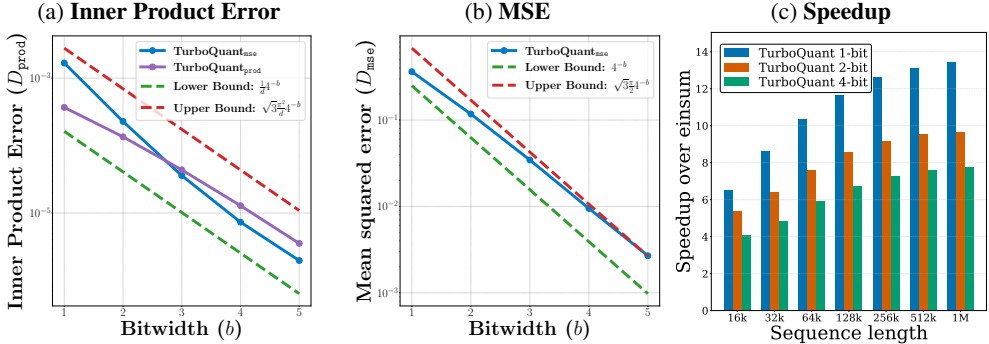

Figure 2: (a) Inner-product error, (b) mean squared error (MSE) versus theoretical bounds, and (c) speedup factors for $QK^\top$ computation in the KV-cache at different bit-widths. Speedup is measured relative to the PyTorch `einsum` baseline.

## 2.2 NEEDLE-IN-A-HAYSTACK

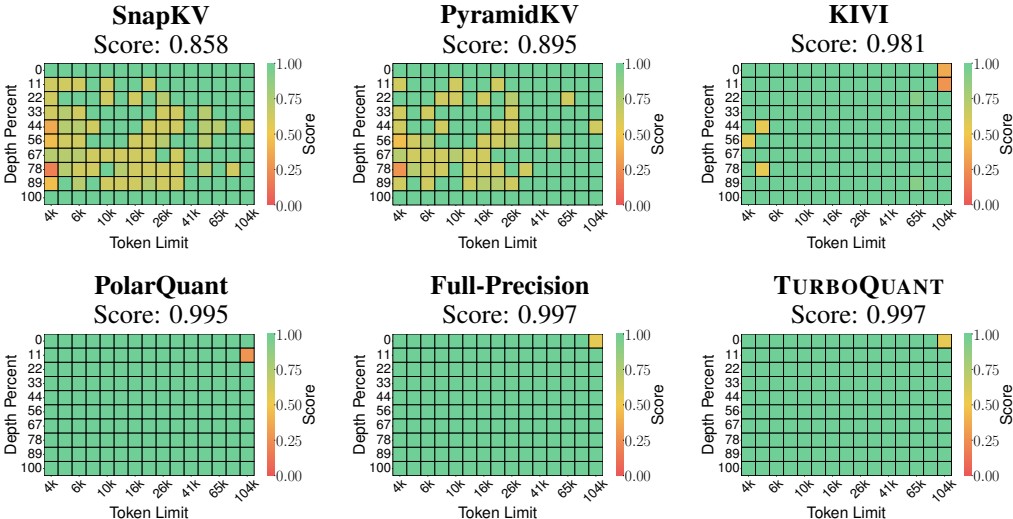

Figure 3: Evaluation of `Llama-3.1-8B-Instruct` on the "Needle-In-A-Haystack" test, where a model must retrieve a hidden sentence from long-context sequences. While some methods struggle with recall, TURBOQUANT, despite being more than $4\times$ quantized, achieves the same exact performance as the uncompressed baseline.

The "Needle-In-A-Haystack Test" Kamradt (2023) evaluates a model's ability to retrieve a unique sentence (the "needle") embedded within a long document (the "haystack"). Following Fu et al. (2024), we perform this test using the `Llama-3.1-8B-Instruct` model across varying document lengths from *4k to 104k tokens*. Performance is measured using the *recall score*, which reflects the accuracy of retrieving the hidden sentence.

We compare our method to state-of-the-art memory-efficient approaches, including PolarQuant Han et al. (2025a), SnapKV Li et al. (2024), PyramidKV Cai et al. (2024), and KIVI Liu et al. (2024b), all evaluated under a memory compression ratio of 0.25 (i.e., using only 25% of the full KV cache). As shown in Fig. 3, quantization methods with theoretical guarantees—such as PolarQuant and TURBOQUANT —outperform token-level compression (SnapKV, PyramidKV) and scalar quantization methods without formal guarantees (KIVI). Remarkably, TURBOQUANT matches the performance of the full-precision model even at $4\times$ compression, highlighting its robustness for long-context tasks.

| Method | KV Size | SingleQA | MultiQA | Summarization | Few shot | Synthetic | Code | Average |
|---|---|---|---|---|---|---|---|---|
| | | | Llama-3.1-8B-Instruct | | | | | |
| Full Cache | 16 | 45.29 | 45.16 | 26.55 | 68.38 | 59.54 | 46.28 | 50.06 |
| KIVI | 3 | 43.38 | 37.99 | 27.16 | 68.38 | 59.50 | 44.68 | 48.50 |
| KIVI | 5 | 45.04 | 45.70 | 26.47 | 68.57 | 59.55 | 46.41 | 50.16 |
| PolarQuant | 3.9 | 45.18 | 44.48 | 26.23 | 68.25 | 60.07 | 45.24 | 49.78 |
| TURBOQUANT (ours) | 2.5 | 44.88 | 44.01 | 26.75 | 68.39 | 59.07 | 46.03 | 49.74 |
| TURBOQUANT (ours) | 3.5 | 45.01 | 45.31 | 26.00 | 68.63 | 59.95 | 46.17 | 50.06 |
| | | | Ministral-7B-Instruct | | | | | |
| Full Cache | 16 | 47.53 | 49.06 | 26.09 | 66.83 | 53.50 | 47.90 | 49.89 |
| TURBOQUANT (ours) | 2.5 | 48.38 | 49.22 | 24.91 | 66.69 | 53.17 | 46.83 | 49.62 |

Table 1: LongBench-V1 Bai et al. (2023) results of various KV cache compression methods on `Llama-3.1-8B-Instruct`.

## 2.3 END-TO-END GENERATION ON LONGBENCH

We evaluated TURBOQUANT on the LongBench dataset Bai et al. (2023), using the more uniformly distributed **LongBench-E** subset to ensure a fair comparison across context lengths. Our method is compared to previous approaches in Section 2.2 using both `Llama-3.1-8B-Instruct` and `Ministral-7B-Instruct`. Unlike **KIVI** and **PolarQuant**, which skip quantization for generated tokens, TURBOQUANT applies quantization throughout the streaming process.

As shown in Table 1, TURBOQUANT consistently outperforms other methods, achieving strong results even under low-precision settings—specifically, **2.5-bit** and **3.5-bit** quantization. These non-integer precisions arise from a two-tier channel-wise quantization strategy: outlier channels are allocated more bits (e.g., 32 channels at 3 bits, 96 at 2 bits for 2.5-bit precision), inspired by prior work Zandieh et al. (2024a); Su et al. (2025). Despite operating at lower bitwidths, TURBOQUANT matches the performance of unquantized baselines while achieving over $4.5\times$ compression.

## 2.4 NEAR NEIGHBOUR SEARCH EXPERIMENTS

In this section, we demonstrate the effectiveness of TURBOQUANT in near-neighbor search tasks. Experiments are conducted on the DBpedia Entities dataset Thakur et al. (2021), encoded using OpenAI3 embeddings in 1536- and 3072-dimensional spaces,... [1] [2] as well as on a lower-dimensional dataset using standard GloVe embeddings Pennington et al. (2014).

For each dataset, we use the full set of database vectors for training and sample 10,000 distinct entries as queries, except for GloVe, where we use the pre-existing 10,000-query set[3]. We compare TURBOQUANT against two baselines: Product Quantization Douze et al. (2024) and RabitQ Gao et al. (2024). We report recall@k, which measures whether the true top inner-product neighbor is recovered within the top-$k$ approximate results.

**Product Quantization (PQ)** Douze et al. (2024) uses k-means to construct codebooks, incurring exponential growth in storage as bit-width increases. For efficient querying, we use AVX2-based implementations with LUT256 (256 codewords), grouping 4 coordinates for 2-bit and 2 coordinates for 4-bit quantization. Although PQ benefits from using the same data for training and evaluation, it still suffers from quality degradation at low-bit LUT16 configurations.

**RabitQ** Gao et al. (2024) lacks vectorized implementation and GPU support, leading to significantly slower CPU performance. Furthermore, its actual bit usage is higher than reported due to hidden computational overheads, which are not included in the bit ratio accounting.

Despite these advantages favoring the baselines, TURBOQUANT attains the highest recall@k in almost all settings across datasets and bit-widths. The only exceptions occur in a few GloVe 2-bit settings, where RabitQ achieves slightly higher recall. Overall, TURBOQUANT remains

---

[1] https://huggingface.co/datasets/Qdrant/dbpedia-entities-openai3-text-embedding-3-large-1536-1M

[2] https://huggingface.co/datasets/Qdrant/dbpedia-entities-openai3-text-embedding-3-large-3072-1M

[3] http://ann-benchmarks.com/glove-200-angular.hdf5

the strongest performer across the remaining settings, demonstrating its robustness for high-dimensional, quantization-based search.

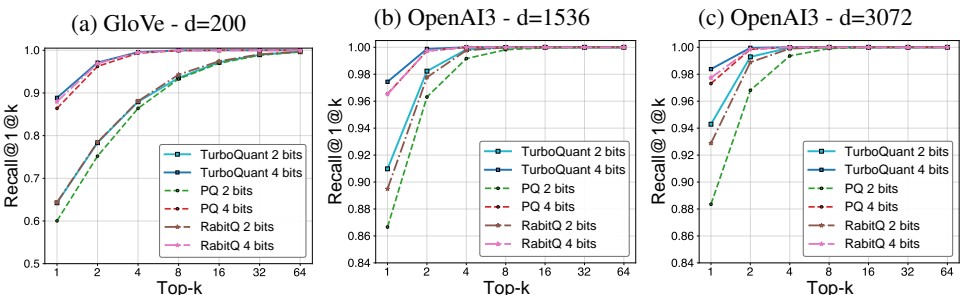

Figure 4: Recall comparison on different datasets with different embedding dimensions.

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

## A   EXTENDED PRIOR WORK

**Beginnings of VQ.**   The vector quantization theory started by Shannon's seminal work Shannon (1948); Shannon et al. (1959) on achievable distortion-rate functions. In 1963, Zador Zador (1964) made significant advances by employing high-resolution methods to derive the limiting operational distortion-rate function for fixed-rate quantization at high rates that closely matches Shannon's distortion-rate function. However, Zador did not specifically consider implementable algorithms. Gersho's influential paper Gersho (1979), further advanced the vector quantization by popularizing high-resolution theory, simplifying Zador's results, introducing lattice vector quantization, and proposing a key conjecture that shaped the field. Despite these theoretical advancements, the practical applicability of vector quantization remained unclear in early years. The most straightforward encoding method, brute-force nearest neighbor search, was computationally expensive, hindering the adoption of VQ in practice.

**Rotation-based Quantization.**   Random orthogonal or Hadamard-style rotations have become a standard and effective preprocessing tool in quantization, especially for reducing outliers and improving coordinate regularity before scalar quantization. Representative examples include Chee et al. (2024); Xi et al. (2023); Ashkboos et al. (2024b). Related methods such as HIGGS Malinovskii et al. (2024) and EDEN Vargaftik et al. (2022) and prior work by Hadad et al. Hadad & Erez (2016) also combine rotation with an MSE-oriented one-dimensional codebook. Our work is similar in spirit, but our main novelty is theoretical: we derive the exact coordinate distribution induced by random rotation and, to the best of our knowledge, provide the first formal proof of optimality for this rotated-coordinate scalar quantization rule.

**Online KV Cache Compression.**   Several approaches have been proposed to compress the KV cache. These include architectural modifications Shazeer (2019); Ainslie et al. (2023); Dai et al. (2024) which restructure the transformer to minimize the number of stored key-value pairs. Additionally, pruning or evicting redundant or less critical tokens has emerged as another approach Beltagy et al. (2020); Zhang et al. (2024b); Liu et al. (2024a); Xiao et al. (2023b); Zandieh et al. (2024b); Li et al. (2024); Han et al. (2025b).

A simple yet effective approach to reducing KV cache size is quantizing the KV cache. Several quantization techniques have been developed specifically for this purpose Yue et al. (2024); Yang et al. (2024); Dong et al. (2024); Kang et al. (2024); Zhang et al. (2024a); Liu et al. (2024b); Hooper et al. (2024); Kim et al. (2024); Han et al. (2025a). Recently, a new quantization called QJL Zandieh et al. (2024a) introduced an efficient, data-oblivious 1-bit quantization approach based on sketching techniques, which provides unbiased estimates for inner product queries. This method does not require tuning or adaptation to the input data and we make use of this technology in our quantizer optimized for inner product distortion.

**Product Quantization (PQ).**   In Near Neighbor (NN) search problem with Euclidean datasets, the index size poses a significant memory bottleneck, often mitigated by quantization techniques, commonly referred to as Product Quantization (PQ) in the NN literature. Many of these algorithms rely on constructing a quantization codebook using variations of k-means during the indexing phase Jegou et al. (2010); Babenko & Lempitsky (2014); Ge et al. (2013); Wang et al. (2017); Guo et al. (2020). Therefore, these methods are ill-suited for online settings due to their requirement for extensive preprocessing.

**Explicit Constants vs. Asymptotic Optimality.**   Recent work of Gao et al. (2024) proposed a grid-based quantization method that avoids preprocessing by projecting a uniform grid onto the unit sphere and assigning each data point to its nearest projection; their analysis establishes asymptotic optimality of required bitwidth for a desired distortion up to hidden constants. Our contribution is giving an explicit characterization of the distortion rate as a function of the bitwidth. This form makes clear that not only the asymptotic scaling, but also the constant factor in the bit budget, is important in practice. Moreover, our quantization procedure is naturally aligned with vectorized execution, which makes efficient GPU implementation straightforward.

# B  MATHEMATICAL PRELIMINARIES

For a random variable $x$ we denote its differential entropy as $h(x)$. For random variables $x$ and $y$, the mutual information between them is denoted as $I(x; y) = h(x) - h(x|y)$.

Given that TURBOQUANT employs random rotation to mitigate worst-case input scenarios, understanding the statistical properties of random points on a hypersphere is essential. The following lemma outlines one such property that we will need for analysis and design purposes:

**Lemma 1** (coordinate distribution of random point on hypersphere). *For any positive integer $d$ if $\boldsymbol{x} \in \mathbb{S}^{d-1}$ is a random variable uniformly distributed over the unit hypersphere, then for any $j \in [d]$ the coordinate $\boldsymbol{x}_j$ follows the following (scaled/shifted) Beta distribution:*

$$\boldsymbol{x}_j \sim f_X(x) := \frac{\Gamma(d/2)}{\sqrt{\pi} \cdot \Gamma((d-1)/2)} \left(1 - x^2\right)^{(d-3)/2}.$$

*In high dimensions this beta distribtion converges to the normal distribution $f_X(\cdot) \to \mathcal{N}(0, 1/d)$.*

*Proof.* $f_X(x)$ equals the ratio of the area of a sphere with radius $\sqrt{1 - x^2}$ in dimension $d - 1$ to the volume of a unit sphere in dimension $d$ scaled down by $1/\sqrt{1 - x^2}$ (by Pythagorean theorem). Therefore,

$$f_X(x) = \frac{\frac{2\pi^{(d-1)/2}}{\Gamma((d-1)/2)} \cdot (1 - x^2)^{(d-2)/2}}{\frac{2\pi^{d/2}}{\Gamma(d/2)}} \cdot 1/\sqrt{1 - x^2} = \frac{\Gamma(d/2)}{\sqrt{\pi} \cdot \Gamma((d-1)/2)} \left(1 - x^2\right)^{(d-3)/2}.$$

$\square$

## B.1  SHANNON LOWER BOUND ON DISTORTION

The Shannon Lower Bound (SLB) is a powerful tool, derived from Shannon's lossy source coding theorem Shannon et al. (1959), that provides a universal lower bound on the optimal achievable distortion rate for any lossy compression scheme. Specifically, we use a version of SLB tailored for the mean-squared error (MSE) distortion measure applied to general $d$-dimensional sources.

**Lemma 2** (SLB). *Let $\boldsymbol{x} \in \mathbb{R}^d$ be a random vector with an arbitrary probability distribution $p_X$ and finite differential entropy $h(\boldsymbol{x})$. Define the MSE distortion-rate function $D(B)$ for total bit complexity $B \geq 0$ as:*

$$D(p_X, B) := \inf \left\{ \mathbb{E}\left[ \|\boldsymbol{x} - \boldsymbol{y}\|_2^2 \right] : I(\boldsymbol{x}; \boldsymbol{y}) \leq B \right\},$$

*where the infimum is taken over all joint distributions of $\boldsymbol{x}$ and a reconstruction random vector $\boldsymbol{y} \in \mathbb{R}^d$ such that the mutual information $I(\boldsymbol{x}; \boldsymbol{y})$ is at most $B$ and $\mathbb{E}\left[ \|\boldsymbol{x} - \boldsymbol{y}\|_2^2 \right]$ is the expected MSE distortion, calculated with respect to the joint distribution of $\boldsymbol{x}$ and $\boldsymbol{y}$. Then, for any bit complexity $B \geq 0$, the following Shannon Lower Bound holds:*

$$D(p_X, B) \geq \frac{d}{2\pi e} \cdot 2^{(2/d)(h(\boldsymbol{x}) - B)}.$$

This is a classic result proved using backward Gaussian test channel (for a proof see Cover (1999)). Our lower bound result uses a corollary of SLB that corresponds to the uniformly distributed random points on the unit hyeprsphere. We present this in the following lemma:

**Lemma 3** (SLB for random point on hypersphere). *Let $\boldsymbol{x} \in \mathbb{S}^{d-1}$ be a random variable uniformly distributed over the unit hypersphere and define the MSE distortion-rate function $D(B)$ for total bit complexity $B$ as per Lemma 2. Then, for any bit complexity $B \geq 0$, the following distortion lower bound holds:*

$$D(B) \geq 2^{-2B/d}.$$

*Proof.* If we let $A_d$ denote the area of the hypersphere $\mathbb{S}^{d-1}$, the entropy of uniform distribution over hypersphere is $h(\boldsymbol{x}) = \log_2 A_d$. Plugging this into the SLB from Lemma 2 we get $D(B) \geq \frac{d}{2\pi e} \cdot A_d^{2/d} \cdot 2^{-2B/d}$. Using Stirling's approximation formula for Gamma function we have $A_d = \frac{2\pi^{d/2}}{\Gamma(d/2)} \geq \left(\frac{2\pi e}{d}\right)^{d/2} \cdot \sqrt{\frac{2d}{\pi}} \cdot (1 - O(1/d))$. By substituting this into the inequality obtained from Lemma 2 we get the desired lower bound. $\square$

## B.2 QJL: 1-BIT INNER PRODUCT QUANTIZATION

As previously stated, we design two VQ algorithms: one optimized for minimizing MSE and the other for minimizing inner product error. We show that MSE-optimal quantizers do not necessarily provide unbiased inner product estimates, particularly exhibiting significant bias at lower bit-widths. Our solution for inner product quantization is a two-stage algorithm. First, we apply the MSE-optimal quantizer using one less bit than the desired bit-width budget, thus minimizing the L2 norm of the residuals. Next we apply an unbiased and optimal single-bit quantizer to the residual. For the single-bit inner product quantizer, we utilize the recently proposed Quantized Johnson-Lindenstrauss (QJL) algorithm Zandieh et al. (2024a), which is an optimal inner product quantizer with a bit-width of one. Here, we present the QJL algorithm and its essential theoretical guarantees.

**Definition 1** (QJL). *For any positive integer $d$ the QJL map $Q_{\mathtt{qjl}} : \mathbb{R}^d \to \{-1, +1\}^d$ is defined as:*

$$Q_{\mathtt{qjl}}(\boldsymbol{x}) := \mathtt{sign}\left(\boldsymbol{S} \cdot \boldsymbol{x}\right) \quad \text{for any } \boldsymbol{x} \in \mathbb{R}^d,$$

*where $\boldsymbol{S} \in \mathbb{R}^{d \times d}$ is a random matrix with i.i.d. entries sampled from the normal distribution $\mathcal{N}(0, 1)$ and the $\mathtt{sign}$ function is applied entry-wise to its vector input. The inverse/dequantization map $Q_{\mathtt{qjl}}^{-1} : \{-1, +1\}^d \to \mathbb{R}^d$ is defined as:*

$$Q_{\mathtt{qjl}}^{-1}(\boldsymbol{z}) := \frac{\sqrt{\pi/2}}{d} \cdot \boldsymbol{S}^\top \cdot \boldsymbol{z} \quad \text{for any } \boldsymbol{z} \in \{-1, +1\}^d.$$

In the next lemma we restate the results from Zandieh et al. (2024a) that show the QJL is unbiased and also has small inner product distortion:

**Lemma 4** (performance guarantee: QJL). *Let $Q_{\mathtt{qjl}}$ and $Q_{\mathtt{qjl}}^{-1}$ be defined as per Definition 1. For any vector $\boldsymbol{x} \in \mathbb{S}^{d-1}$ and any $\boldsymbol{y} \in \mathbb{R}^d$ we have the following:*

- *Unbiased:* $\mathbb{E}\left[\langle \boldsymbol{y}, Q_{\mathtt{qjl}}^{-1}(Q_{\mathtt{qjl}}(\boldsymbol{x})) \rangle\right] = \langle \boldsymbol{y}, \boldsymbol{x} \rangle.$

- *Variance Bound:* $\mathtt{Var}\left(\langle \boldsymbol{y}, Q_{\mathtt{qjl}}^{-1}(Q_{\mathtt{qjl}}(\boldsymbol{x})) \rangle\right) \le \frac{\pi}{2d} \cdot \|\boldsymbol{y}\|_2^2$

*Proof.* The unbiasedness immediately follows from Lemma 3.2 of Zandieh et al. (2024a). To show the variance bound let $\boldsymbol{s}_1, \boldsymbol{s}_2, \dots \boldsymbol{s}_m$ denote the rows of the random matrix $\boldsymbol{S}$ in Definition 1. We have:

$$\langle \boldsymbol{y}, Q_{\mathtt{qjl}}^{-1}(Q_{\mathtt{qjl}}(\boldsymbol{x})) \rangle = \frac{1}{d} \sum_{i \in [d]} \sqrt{\pi/2} \cdot \boldsymbol{s}_i^\top \boldsymbol{y} \cdot \mathtt{sign}(\boldsymbol{s}_i^\top \boldsymbol{x}).$$

Since $\boldsymbol{s}_i$'s are i.i.d. the above is indeed the average of $d$ i.i.d. random samples defined as $z_i := \sqrt{\pi/2} \cdot \boldsymbol{s}_i^\top \boldsymbol{y} \cdot \mathtt{sign}(\boldsymbol{s}_i^\top \boldsymbol{x})$ for $i \in [d]$. Let us now upper bound the variance of a single $z_i$ using Fact 3.4 from Zandieh et al. (2024a):

$$\mathtt{Var}(z_i) = \pi/2 \cdot \mathtt{Var}\left(\boldsymbol{s}_i^\top \boldsymbol{y} \cdot \mathtt{sign}(\boldsymbol{s}_i^\top \boldsymbol{x})\right) \le \pi/2 \cdot \mathbb{E}\left[(\boldsymbol{s}_i^\top \boldsymbol{y})^2\right] = \pi/2 \cdot \|y\|_2^2, \quad (4)$$

where the last equality above follows because $\boldsymbol{s}_i^\top \boldsymbol{y}$ is a Gaussian random variable with mean zero and variance $\|\boldsymbol{y}\|_2^2$. Now the variance of the average of $d$ i.i.d. random samples $z_1, z_2, \dots z_d$ is:

$$\mathtt{Var}\left(\langle \boldsymbol{y}, Q_{\mathtt{qjl}}^{-1}(Q_{\mathtt{qjl}}(\boldsymbol{x})) \rangle\right) = \frac{1}{d^2} \sum_{i \in [d]} \mathtt{Var}(z_i) \le \frac{\pi}{2d} \cdot \|\boldsymbol{y}\|_2^2.$$

$\square$

## C PROOFS AND TECHNICAL REMARKS

In this section, we provide detailed proofs for each of the main theoretical results presented in the paper, including those for TURBOQUANT$_{\mathtt{mse}}$, TURBOQUANT$_{\mathtt{prod}}$, and the information-theoretic lower bound.

## C.1 TURBOQUANT$_{\texttt{mse}}$

**Remark 1** (Entropy-Encoding Codebook Pointers). *TURBOQUANT's efficiency can be further increased by applying entropy encoding to the indices that point to the closest codebook elements. Specifically, the probability of each codeword index appearing in the quantized vectors can be computed as $p_\ell := \int_{\frac{c_{\ell-1}+c_\ell}{2}}^{\frac{c_\ell+c_{\ell+1}}{2}} f_X(x)\, dx$. Optimally coding the indices, reduces the average bit-width to nearly the entropy of the distribution $\{p_i\}_{i\in[2^b]}$. This lossless compression does not affect the distortion and provides a bit-width reduction at no cost. The most significant reduction occurs for $b = 4$, where the entropy of $\{p_i\}_{i\in[2^b]}$ is approximately $3.8$. Detailed calculations for optimal prefix codes reveal that the average bit-width can be reduced by $5\%$. However, given the limited gain, we have chosen not to incorporate this technique into TURBOQUANT to maintain simplicity and speed.*

**Theorem 1** (Performance Guarantee: TURBOQUANT$_{\texttt{mse}}$). *For any bit-width $b \geq 1$ and any vector $\boldsymbol{x} \in \mathbb{S}^{d-1}$, the procedure QUANT$_{\texttt{mse}}(\boldsymbol{x})$ in Algorithm 1 outputs an index vector $\texttt{idx} \in [2^b]^d$. When this index vector is passed to the primitive DEQUANT$_{\texttt{mse}}(\texttt{idx})$, it produces a reconstructed vector $\tilde{\boldsymbol{x}} \in \mathbb{R}^d$ that satisfies the following distortion bounds:*

- *MSE defined as $D_{\texttt{mse}} := \mathbb{E}_{\tilde{\boldsymbol{x}}}[\|\boldsymbol{x} - \tilde{\boldsymbol{x}}\|_2^2]$ is bounded by $D_{\texttt{mse}} \leq \frac{\sqrt{3}\pi}{2} \cdot \frac{1}{4^b}$ for any $b \geq 0$.*

- *For small bit-widths, specifically $b = 1, 2, 3, 4$ the MSE exhibits finer-grained distortion values: $D_{\texttt{mse}} \approx \mathbf{0.36}, \mathbf{0.117}, \mathbf{0.03}, \mathbf{0.009}$, respectively.*

*Proof.* We start the proof by showing that $D_{\texttt{mse}} = d \cdot \mathcal{C}(f_X, b)$, where $\mathcal{C}(f_X, b)$ is the optimal MSE cost for scalar quantizer defined in Eq. (3). Let $\tilde{\boldsymbol{y}}$ be defined as per line 9 of Algorithm 1. Since $\boldsymbol{\Pi}$ is a rotation matrix we can write: $\|\boldsymbol{x} - \tilde{\boldsymbol{x}}\|_2 = \|\boldsymbol{\Pi} \cdot \boldsymbol{x} - \tilde{\boldsymbol{y}}\|_2$. Using the notation $\boldsymbol{y} = \boldsymbol{\Pi} \cdot \boldsymbol{x}$ as per line 5 of Algorithm 1 and plugging this into the definition of $D_{\texttt{mse}}$ we can write:

$$
\begin{aligned}
D_{\texttt{mse}} &= \mathbb{E}[\|\boldsymbol{y} - \tilde{\boldsymbol{y}}\|_2^2] \\
&= \sum_{j\in[d]} \mathbb{E}\left[|\boldsymbol{y}_j - \tilde{\boldsymbol{y}}_j|^2\right] \\
&= \sum_{j\in[d]} \mathbb{E}\left[|\boldsymbol{y}_j - c_{\texttt{idx}_j}|^2\right] \\
&= d \cdot \mathbb{E}\left[|\boldsymbol{y}_1 - c_{\texttt{idx}_1}|^2\right] \\
&= d \cdot \min_{-1 \leq c_1 \leq c_2 \leq \dots \leq c_{2^b} \leq 1} \sum_{i=1}^{2^b} \int_{\frac{c_{i-1}+c_i}{2}}^{\frac{c_i+c_{i+1}}{2}} |x - c_i|^2 \cdot f_X(x)\, dx \\
&= d \cdot \mathcal{C}(f_X, b).
\end{aligned}
$$

The third equality above follows from the definition of $\tilde{\boldsymbol{y}}$ in line 9 of Algorithm 1 and the fourth line above follows because all $\boldsymbol{y}_j$'s have identical distribution of $\boldsymbol{y}_j \sim f_X(\cdot)$ as shown in Lemma 1. The last two lines above follows because $c_{\texttt{idx}_j}$ is chosen to be the nearest centroid to each coordinate $\boldsymbol{y}_j$ in line 6.

Now we must bound the optimal k-means cost $\mathcal{C}(f_X, b)$. For moderate values of $d$, $f_X \to \mathcal{N}(0, 1/d)$. By numerically solving the optimization problem in Eq. (3) for values $b = 1, 2, 3, 4$ we get that $\mathcal{C}(f_X, b) \approx \frac{0.36}{d}, \frac{0.117}{d}, \frac{0.03}{d}, \frac{0.009}{d}$, respectively. For larger bit-widths $b > 4$, we can apply the Panter-Dite Panter & Dite (1951) high-resolution formula for the distortion of a fixed-rate scalar quantizer, yielding the following bound:

$$
\mathcal{C}(f_X, b) \leq \frac{1}{12} \cdot \left(\int f_X(x)^{1/3}\, dx\right)^3 \cdot \frac{1}{4^b} = \frac{\sqrt{3}\pi}{2d} \cdot \frac{1}{4^b}.
$$

This completes the proof. $\square$

## C.2 TURBOQUANT$_{\texttt{prod}}$

**Theorem 2** (Performance Guarantee: TURBOQUANT$_{\texttt{prod}}$). *For any bit-width $b \geq 1$ and any vector $\boldsymbol{x} \in \mathbb{S}^{d-1}$, the procedure QUANT$_{\texttt{prod}}(\boldsymbol{x})$ in Algorithm 2 outputs an index vector $\texttt{idx} \in [2^{b-1}]^d$*

*along with a sign vector* $\mathtt{qjl} \in \{-1, 1\}^d$ *and a positive number* $\gamma \geq 0$. *When these vectors and the scalar value are passed to the primitive* $\mathrm{DEQUANT}_{\mathrm{prod}}(\mathtt{idx}, \mathtt{qjl}, \gamma)$, *it produces a reconstructed vector* $\tilde{\boldsymbol{x}} \in \mathbb{R}^d$ *that for any vector* $\boldsymbol{y} \in \mathbb{R}^d$ *satisfies the following properties:*

- *Expected inner-product* $\mathbb{E}_{\tilde{\boldsymbol{x}}}[\langle \boldsymbol{y}, \tilde{\boldsymbol{x}} \rangle] = \langle \boldsymbol{y}, \boldsymbol{x} \rangle$

- *Inner-product distortion defined as* $D_{\mathrm{prod}} := \mathbb{E}_{\tilde{\boldsymbol{x}}}\left[ |\langle \boldsymbol{y}, \boldsymbol{x} \rangle - \langle \boldsymbol{y}, \tilde{\boldsymbol{x}} \rangle|^2 \right]$ *is bounded by* $D_{\mathrm{prod}} \leq \frac{\sqrt{3}\pi^2 \cdot \|\boldsymbol{y}\|_2^2}{d} \cdot \frac{1}{4^b}$ *for any* $b \geq 0$.

- *For small bit-widths, specifically* $b = 1, 2, 3, 4$, $D_{\mathrm{prod}}$ *exhibits finer-grained distortion values:* $D_{\mathrm{prod}} \approx \frac{1.57}{d}, \frac{0.56}{d}, \frac{0.18}{d}, \frac{0.047}{d}$, *respectively.*

*Proof.* First we compute the conditional expectation of the inner product estimate $\langle \boldsymbol{y}, \tilde{\boldsymbol{x}} \rangle$ conditioned on $\tilde{\boldsymbol{x}}_{\mathrm{mse}}$ as follows:

$$
\begin{aligned}
\mathbb{E}\left[\langle \boldsymbol{y}, \tilde{\boldsymbol{x}} \rangle | \tilde{\boldsymbol{x}}_{\mathrm{mse}}\right] &= \mathop{\mathbb{E}}_{\tilde{\boldsymbol{x}}_{\mathrm{qjl}}}\left[\langle \boldsymbol{y}, \tilde{\boldsymbol{x}}_{\mathrm{mse}} + \tilde{\boldsymbol{x}}_{\mathrm{qjl}} \rangle | \tilde{\boldsymbol{x}}_{\mathrm{mse}}\right] \\
&= \langle \boldsymbol{y}, \tilde{\boldsymbol{x}}_{\mathrm{mse}} \rangle + \mathop{\mathbb{E}}_{\tilde{\boldsymbol{x}}_{\mathrm{qjl}}}\left[\langle \boldsymbol{y}, \tilde{\boldsymbol{x}}_{\mathrm{qjl}} \rangle | \tilde{\boldsymbol{x}}_{\mathrm{mse}}\right] \\
&= \langle \boldsymbol{y}, \tilde{\boldsymbol{x}}_{\mathrm{mse}} \rangle + \langle \boldsymbol{y}, \boldsymbol{r} \rangle \\
&= \langle \boldsymbol{y}, \boldsymbol{x} \rangle,
\end{aligned}
$$

where the first equality follows from the definition of $\tilde{\boldsymbol{x}}$ in line 12 of the algorithm. The third equality above follows from Lemma 4 and last line follows from definition of the residual vector $\boldsymbol{r} = \boldsymbol{x} - \tilde{\boldsymbol{x}}_{\mathrm{mse}}$ in line 6. Now we can computed the unconditional expectation using the law of total expectation: $\mathbb{E}_{\tilde{\boldsymbol{x}}}[\langle \boldsymbol{y}, \tilde{\boldsymbol{x}} \rangle] = \mathbb{E}_{\tilde{\boldsymbol{x}}_{\mathrm{mse}}}[\mathbb{E}[\langle \boldsymbol{y}, \tilde{\boldsymbol{x}} \rangle | \tilde{\boldsymbol{x}}_{\mathrm{mse}}]] = \mathbb{E}[\langle \boldsymbol{y}, \boldsymbol{x} \rangle] = \langle \boldsymbol{y}, \boldsymbol{x} \rangle$, which proves the first claim of the theorem.

We apply the same conditioning on $\tilde{\boldsymbol{x}}_{\mathrm{mse}}$, when computing the distortion, and then compute the resulting conditional distortion:

$$
\begin{aligned}
\mathbb{E}\left[ |\langle \boldsymbol{y}, \boldsymbol{x} \rangle - \langle \boldsymbol{y}, \tilde{\boldsymbol{x}} \rangle|^2 \,\middle|\, \tilde{\boldsymbol{x}}_{\mathrm{mse}} \right] &= \mathop{\mathbb{E}}_{\tilde{\boldsymbol{x}}_{\mathrm{qjl}}}\left[ |\langle \boldsymbol{y}, \boldsymbol{x} \rangle - \langle \boldsymbol{y}, \tilde{\boldsymbol{x}}_{\mathrm{mse}} + \tilde{\boldsymbol{x}}_{\mathrm{qjl}} \rangle|^2 \,\middle|\, \tilde{\boldsymbol{x}}_{\mathrm{mse}} \right] \\
&= \mathop{\mathbb{E}}_{\tilde{\boldsymbol{x}}_{\mathrm{qjl}}}\left[ |\langle \boldsymbol{y}, \boldsymbol{r} \rangle - \langle \boldsymbol{y}, \tilde{\boldsymbol{x}}_{\mathrm{qjl}} \rangle|^2 \,\middle|\, \tilde{\boldsymbol{x}}_{\mathrm{mse}} \right] \\
&= \mathrm{Var}\left( \langle \boldsymbol{y}, \tilde{\boldsymbol{x}}_{\mathrm{qjl}} \rangle \,\middle|\, \tilde{\boldsymbol{x}}_{\mathrm{mse}} \right) \\
&\leq \frac{\pi}{2d} \cdot \|\boldsymbol{r}\|_2^2 \|\boldsymbol{y}\|_2^2,
\end{aligned}
$$

where the second equality above follows from the definitions of $\boldsymbol{r}$ and $\tilde{\boldsymbol{x}}_{\mathrm{mse}}$ in lines 6 and 10 of Algorithm 2. The third line above follows because $\mathbb{E}[\langle \boldsymbol{y}, \tilde{\boldsymbol{x}}_{\mathrm{qjl}} \rangle] = \langle \boldsymbol{y}, \boldsymbol{r} \rangle$, by Lemma 4. The last line follows from the variance bound of QJL estimator shown in Lemma 4 and using the fact that $\tilde{\boldsymbol{x}}_{\mathrm{qjl}}$ in line 11 is re-scaled by $\gamma = \|\boldsymbol{r}\|$.

Now by law of total expectation along with the fact that $\boldsymbol{r} = \boldsymbol{x} - \tilde{\boldsymbol{x}}_{\mathrm{mse}}$ we can bound the inner product distortion as follows:

$$
\begin{aligned}
D_{\mathrm{prod}} &= \mathop{\mathbb{E}}_{\tilde{\boldsymbol{x}}_{\mathrm{mse}}}\left[ \mathbb{E}\left[ |\langle \boldsymbol{y}, \boldsymbol{x} \rangle - \langle \boldsymbol{y}, \tilde{\boldsymbol{x}} \rangle|^2 \,\middle|\, \tilde{\boldsymbol{x}}_{\mathrm{mse}} \right] \right] \\
&\leq \frac{\pi}{2d} \cdot \|\boldsymbol{y}\|_2^2 \cdot \mathbb{E}[\|\boldsymbol{x} - \tilde{\boldsymbol{x}}_{\mathrm{mse}}\|_2^2] \\
&= \frac{\pi}{2d} \cdot \|\boldsymbol{y}\|_2^2 \cdot D_{\mathrm{mse}}.
\end{aligned}
$$

The theorem follows by invoking the MSE bounds from Theorem 1 with bit-width $b - 1$. $\qquad \square$

## C.3 LOWER BOUNDS

We show that TURBOQUANT achieves an optimal distortion rate, up to a small constant factor, for any bit-width by proving lower bounds on the best achievable distortion for any compression

algorithm. Our lower bound proof leverages Yao's minimax principle. This principle allows us to relate the lower bound for randomized algorithms with worst-case deterministic input vectors to the lower bound for deterministic algorithms with randomized input vectors. Subsequently, we derive a lower bound on the achievable distortion rate for the latter using Shannon's lower bound (SLB) presented in Appendix B.1. Formally, we prove the following theorem.

**Theorem 3** (Lower Bound on Best Achievable Compression Distortion)**.** *For any randomized quantization algorithm* $Q : \mathbb{S}^{d-1} \to \{0,1\}^{b \cdot d}$ *with bit-width* $b$ *and any reconstruction map* $Q^{-1} : \{0,1\}^{b \cdot d} \to \mathbb{R}^d$, *there exist a hard input instance* $\boldsymbol{x} \in \mathbb{S}^{d-1}$ *such that:*

$$D_{\mathtt{mse}}(Q) := \mathbb{E}\left[\left\|\boldsymbol{x} - Q^{-1}\left(Q(\boldsymbol{x})\right)\right\|_2^2\right] \geq \frac{1}{4^b}.$$

*Furthermore, there exists a* $\boldsymbol{y} \in \mathbb{S}^{d-1}$ *such that:*

$$D_{\mathtt{prod}}(Q) = \mathbb{E}\left[\left|\langle \boldsymbol{y}, \boldsymbol{x}\rangle - \langle \boldsymbol{y}, Q^{-1}\left(Q(\boldsymbol{x})\right)\rangle\right|^2\right] \geq \frac{1}{d} \cdot \frac{1}{4^b}$$

We note that a comparable lower bound for the *worst-case* distortion in vector quantization can be derived using "sphere packing" arguments (indeed, with larger constants as this is a harder problem) Gersho (1982). However, Theorem 3 offers a more robust and relevant lower bound for our analysis. This is because it establishes a lower bound on the *expected distortion*, rather than the worst-case error, and aligns seamlessly with our upper bounds presented in Theorem 1 and Theorem 2.

*Proof.* By Yao's minimax principle the expected MSE of the optimal randomized compression algorithm for worst-case inputs ($D_{\mathtt{mse}}$) is equal to the expected MSE of the optimal deterministic compression algorithm when applied to inputs drawn from a maximally difficult randomized distribution. By definition, the MSE of the latter scenario is lower-bounded by the best achievable MSE for inputs uniformly distributed on the unit hypersphere.

The best achievable MSE for a compression algorithm with bit-width $b$, operating on uniformly distributed inputs from the sphere $\mathbb{S}^{d-1}$, is lower bounded in Lemma 3. Therefore, by invoking Lemma 3 we conclude that $D_{\mathtt{mse}} \geq \frac{1}{4^b}$.

Furthermore, from $D_{\mathtt{mse}} \geq \frac{1}{4^b}$ and using the definition of $D_{\mathtt{mse}}$ we conclude that:

$$D_{\mathtt{mse}} = \sum_{j=1}^{d} \mathbb{E}\left[\left|\boldsymbol{x}_j - \left[Q^{-1}\left(Q(\boldsymbol{x})\right)\right]_j\right|^2\right]$$

$$= \sum_{j=1}^{d} \mathbb{E}\left[\left|\langle \boldsymbol{e}_j, \boldsymbol{x}\rangle - \langle \boldsymbol{e}_j, Q^{-1}\left(Q(\boldsymbol{x})\right)\rangle\right|^2\right]$$

$$\geq \frac{1}{4^b}.$$

By pigeonhole principle there exist an index $j \in [d]$ such that $\mathbb{E}\left[\left|\langle \boldsymbol{e}_j, \boldsymbol{x}\rangle - \langle \boldsymbol{e}_j, Q^{-1}\left(Q(\boldsymbol{x})\right)\rangle\right|^2\right] \geq \frac{1}{d} \cdot \frac{1}{4^b}$, which completes the proof. $\square$

## D  ADDITIONAL EXPERIMENT

As observed in Fig. 5, when quantizing to 2 bits, the variance remains constant regardless of the inner product of the original vector in the TURBOQUANTprod approach. However, the same plot indicates that the bias in the TURBOQUANTmse approach is dependent on the average inner product. As the average inner product increases, the bias also increases.

## E  IMPLEMENTATION

The TURBOQUANT algorithm, as detailed in Algorithm 1 (for MSE-optimal quantization) and Algorithm 2 (for inner product quantization), incorporates random rotation matrices applied to the

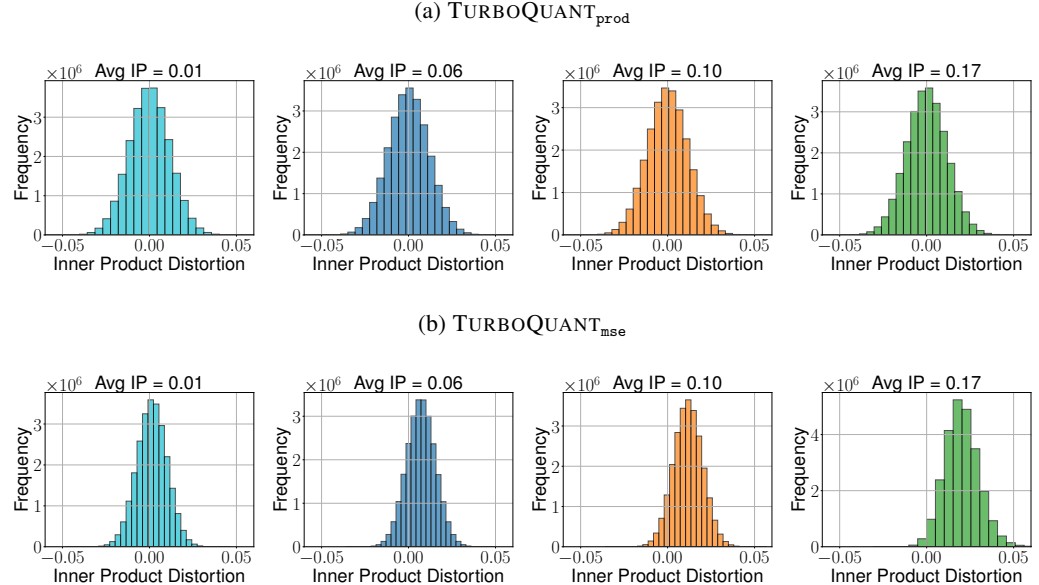

Figure 5: The variance of Inner-product error remains constant for TURBOQUANT$_{\text{prod}}$, while in TURBOQUANT$_{\text{mse}}$ increases with the average inner product. Bit-width is $b = 2$.

input embedding vectors during both quantization and dequantization stages. This transformation is crucial for the algorithm's performance. In practice, the rotation matrices are implemented as structured matrices, specifically *randomized Hadamard transforms* Ahle et al. (2020). This involves randomly flipping the signs of the embedding vector coordinates, followed by the fast Hadamard transform (FHT), enabling matrix-vector multiplication in $O(d \log d)$ time, a significant improvement over the naive $O(d^2)$ complexity. This efficient transformation is beneficial for both CPU and GPU architectures.

The overall quantization process $Q_{\text{prod}}$ with bit-width $b$ can be conceptualized in two stages. First, TURBOQUANT$_{\text{mse}}$ ($Q_{\text{mse}}$) with bit-width $b-1$ is applied to the input vector $\boldsymbol{x}$. Second, the residual error of this stage, $\boldsymbol{r}$, is then quantized using a 1-bit Quantized Johnson-Lindenstrauss (QJL) transform, denoted as $\text{qjl}(\boldsymbol{r})$. The final quantized representation $\tilde{\boldsymbol{x}}$ stores the necessary information from both $Q_{\text{mse}}(\boldsymbol{x})$ and $\text{qjl}(\boldsymbol{r})$.

**Vector Search.** For efficient search over a vector dataset $\boldsymbol{x}_1, \ldots \boldsymbol{x}_n \in \mathbb{R}^d$, vectors are quantized using TURBOQUANT$_{\text{prod}}$ to yield $\tilde{\boldsymbol{x}}_1, \ldots \tilde{\boldsymbol{x}}_n$. Given a query $\boldsymbol{q} \in \mathbb{R}^d$, we want to estimate the inner products $\langle \boldsymbol{q}, \boldsymbol{x}_i \rangle$ through computing $\langle \boldsymbol{q}, Q_{\text{prod}}^{-1}(\tilde{\boldsymbol{x}}_i) \rangle$. By dequant primitives in Algorithm 1 and Algorithm 2 we have $\langle \boldsymbol{q}, Q_{\text{prod}}^{-1}(\tilde{\boldsymbol{x}}_i) \rangle = \|\text{qjl}(\boldsymbol{r}_i)\| \cdot \langle \boldsymbol{Sq}, \text{qjl}(\boldsymbol{r}_i) \rangle + \langle \boldsymbol{\Pi q}, c_{Q_{\text{mse}}(\boldsymbol{x}_i)} \rangle$, where $c_j$'s are the centroids that minimize the objective in Eq. (3) and $\boldsymbol{Sq}$ and $\boldsymbol{\Pi q}$ are randomized Hadamard transforms applied on $\boldsymbol{q}$ which can be computed very quickly.

Computing the first inner product reduces to accumulating the coordinates of $\boldsymbol{Sq}$ with signs given in the quantized vector $\text{qjl}(\boldsymbol{r}_i)$. The second inner product, $\langle \boldsymbol{\Pi q}, Q_{\text{mse}}(\boldsymbol{x}_i) \rangle$, can also be computed with high efficiency. This computation involves summing contributions from quantized components of $\boldsymbol{x}_i$. To accelerate this, for each query $\boldsymbol{q}$, we precompute $d$ small lookup tables (LUTs). For the $k$-th component, its LUT, $\boldsymbol{L}^{(k)}$, would store values $\boldsymbol{L}_j^{(k)} = (\boldsymbol{\Pi q})_k \cdot c_j$, where $(\boldsymbol{\Pi q})_k$ is the $k$-th coordinate of the transformed query. This can be implemented efficiently using the "LUT16" AVX2 in-register lookup technique described by Wu et al. (2019); Ge et al. (2013) for any bit-width $b$ at most $b \leq 5$. This results in (at most) 16-entry tables for each component, and the PSHUFB instruction can then perform multiple such lookups in parallel. This involves quantizing the LUT entries themselves (e.g., to 8-bit values) to fully leverage the SIMD capabilities.

**KV Cache.** In LLMs the Key and Value caches $\boldsymbol{K}, \boldsymbol{V} \in \mathbb{R}^{n \times d}$ (where $n$ is the sequence length, and $d$ is the embedding dimension) can be compressed by quantizing their row vectors using $\text{TurboQuant}_{\text{prod}}$ yielding $\tilde{\boldsymbol{K}}$ and $\tilde{\boldsymbol{V}}$. During autoregressive decoding, to compute the attention output for a query vector $\boldsymbol{q} \in \mathbb{R}^d$, we first need to calculate attention scores, typically involving $\boldsymbol{q} \cdot Q_{\text{prod}}^{-1}(\tilde{\boldsymbol{K}})^\top$. These scores, forming a vector $\boldsymbol{a} \in \mathbb{R}^n$ (after softmax), are then used to retrieve values via $\boldsymbol{a} \cdot Q_{\text{prod}}^{-1}(\tilde{\boldsymbol{V}})$.

Instead of explicit dequantization of $\tilde{\boldsymbol{K}}$ and $\tilde{\boldsymbol{V}}$ back to full precision in HBM, a more efficient approach is to use a fused GPU kernel. This kernel performs on-the-fly dequantization and mixed-precision matrix multiplication concurrently. It loads the quantized key or value data from HBM into on-chip shared memory and registers. The dequantization is then performed based on Algorithm 2 (the codebook is stored as a LUT), and the resulting values are immediately used in the matrix multiplication (e.g., GEMM operations with M=1), thereby minimizing costly data transactions with HBM.

Indeed, fast and efficient mixed-precision fused kernels for this type of non-uniform LUT-based quantizers were recently developed in the FLUTE paper Guo et al. (2024), initially for the static weights of LLMs. FLUTE's implementation uses optimized workload distribution (e.g., Stream-K) techniques to maximize efficiency. Our implementation of TurboQuant mixed-precision matrix multiplication for KV cache compression builds upon their publicly available code, with modifications to the dequantization module. Our findings indicate that this mixed-precision fused kernel implementation for KV cache operations is 2-4× faster than conventional floating-point GEMM kernels.

