# OpenReview forum: "TurboQuant: Online Vector Quantization with Near-optimal Distortion Rate"
_ICLR.cc/2026/Conference — ICLR 2026 Poster_

### Official Review · Reviewer_k6Lz · 2025-10-26

**Soundness:** 1
**Presentation:** 3
**Contribution:** 2
**Rating:** 4
**Confidence:** 3

**Summary:**

The author introduce a new method for online vector quantization, which does not assume data-specific tuning, which is especially useful for applications such as KV cache compression in LLMs. Their method consist of applying a random rotation to map the (unit-normalized) distribution to a representation with near-independent coordinates whose distribution converges to a known distribution in high dimensions. This allows them to pre-compute per-coordinate $b$-bits codebooks optimized for this distribution, that are suitable for online usage. They further combine this method with QJL to provide unbiased inner product, which is important for KV cache compression.

**Strengths:**

- The proposed method is clearly explain and simple to use.
- The proposed method is online, making it easily applicable to KV cache compression.
- The authors provide theoretical justifications and bounds.
- The proposed method has potential for broad application on LLM inference

**Weaknesses:**

While the presented method has undeniable strengths, they are undermined by weak experimental validations.
- For KV compression, all compared methods use a different number of bits, making comparison of the accuracy-compression ratio harder to read. Better analysis of where this method takes place on this trade-off is needed.
- KL compression is compared to only two other comparison methods, ignoring some other methods, including RotateKV which is cited by the authors. Comprehensive comparison is needed.
- Authors propose “TurboQuant”, which revolves around the random matrix multiplication followed by per-bit quantization. The QJL and two-tier channel-wise quantization strategy parts come from other work and are generally applicable to some other methods, but are used in the comparison. Analysis of the individual component introduce by the paper would make results more convincing.
- This method is only compared to existing ones on a single benchmark, and single model.

The near neighbor search experiments are the least convincing:
- The method is only compared to PQ, and another method that barely surpasses PQ. For this kind of data-dependant method already surpass these baselines, including codebook-based methods (such as OPQ, RQ, LSQ++) or neural-network based (such as QINCo2 or UNQ).
- An argument is made about the time needed to quantize the training set, but this has very limited impact. Real-world ANN settings are usually constrained mostly by the search speed and accuracy on CPU, and the limited memory, more than the 1-time quantization. This part lacks justification.

**Questions:**

The method is interesting, and its simplicity is a strong argument for applying it. I would ask the authors to clarify their results and clearly justify the quality of the quantization in either KV compression or ANN settings.

---

> ### Author Response · Authors · 2025-11-23
> **Response to the review**
>
> We appreciate the reviewer’s insightful feedback. Our response to you're comments/concerns are as follows:
>
> `Additional KV compression experiments: same bit-widths, RotateKV benchmark, and ablating the impact of two-tier channel-wise quantization strategy`:
>
> First, we'd like to clarify the goal of the KV cache experiments in table 1 was to show TurboQuant is consistently better than models with higher bitwidths, simultaneously reducing the bitwidth and increasing the quality scores. For each competing method we let the bitwidth of that method to be more than TurboQuant and showed that even in this case TurboQuant still has higher accuracy. Please note that this demonstrates the strength of TurboQuant relative to other methods even more than if we compared them at the same number of bits.
>
> Furthermore, we have now run new experiments on the GSM8K mathematical reasoning dataset, using the Qwen2-7B-Instruct model (both model and dataset are new additions not present in the initial submission), fixing the bit-width for all methods at 3 bits. This analysis also incorporates a comparison against the new baseline method, **KV-Rotate** suggested by the reviewer. The resulting performance table is presented below:
> | Qwen2-7B-Instruct                  | GSM8K |
> |-------------------------|--------------:|
> | Exact (FP16) |85.7%       |
> | **TurboQuant (3-bits)**         | **84.3%**       |
> |KV-Rotate (3-bits) | 83.6%       |
> | KIVI (3-bits)       | 81.6%       |
>
> Next, we present the experimental results on the long-bench dataset from Table 1 while the two-tier channel-wise quantization is disabled and we precisely used the TurboQuant algorithm from Algorithm-1 of the paper. The results for 3-bit TurboQuant on Llama-3.1-8B-Instruct model are as follows:
> | Method                      |     SingleQA |     MultiQA |     Summarization |    Few shot |    Synthetic |    Code |    Average |
> |-------------------------|--------------:|-------------:|--------------:|------------------:|----------------:|----------------:|----------------:|
> | TurboQuant (3-bits)  | 44.66         | 44.16        | 25.80         | 68.01             | 59.65           | 45.76           | 49.54         |
>
> While a slight degradation in quality is observed, the resulting performance remains superior to that of other baselines employing comparable bit-widths. We commit to integrating both these results and the complete set of new findings into the final version of the paper.
>
> `ANN experiments with stronger PQ methods`:
>
> Thank you for suggesting new relevant quantization methods to include in our ANN studies. We ran new experiments to benchmark our performance against more recent quantization papers, including NestQuant, LSQ++, and OPQ. For your initial review, we present the recall metrics on the GloVe ($d=200$) dataset with bit-width=4, and we commit to incorporating the results for the OpenAI datasets as well in the final version of the paper.
> | Method                  | Recall@1 | Recall@2 | Recall@4 | Recall@8 | Recall@16 | Recall@32 |
> |-------------------------|--------------:|-------------:|--------------:|------------------:|----------------:|----------------:|
> | **TurboQuant**         | **0.8829**       | **0.9756**      | **0.9978**       | **0.9999**           | 1.0         | 1.0         |
> |NestQuant | 0.8249       | 0.9460      | 0.9917       | 0.9988           | 1.0         | 1.0         |
> | LSQ++       | 0.8587       | 0.9664      | 0.9965       | 0.9993           | 1.0         | 1.0         |
> | OPQ       | 0.8229       | 0.9456      | 0.9956       | 0.9978           | 1.0         | 1.0         |
>
> `on the importance of quantization time`:
>
> We agree with the reviewer that, for the typical Approximate Nearest Neighbor (ANN) problem, the one-time indexing runtime is often less critical than the query or search latency. However, there are important applications requiring online and dynamic ANN—such as certain sparse attention inference servers that rely on real-time ANN to identify top-k attention scores—place a crucial premium on quantization and indexing speed. The primary use case for our online quantization approach is the Key-Value (KV) cache, where data points genuinely arrive as a stream, necessitating rapid, on-the-fly quantization where the temporal cost of quantization is highly significant. The ANN application serves mainly as a demonstration of our quantization’s quality which was able to achieve very high recall.

---

> > ### Comment · Reviewer_k6Lz · 2025-11-24
> >
> > I would like to thank the authors for providing additional results on each aspect and enhancing the soundness of the paper. The results with KV-Rotate and two-tier channel-wise quantization disabled clarify the performance of TurboQuant against other methods for the main results. Adding better baselines for ANN and clarifying potential application for fast data-independent quantization make this part much more sound.
> >
> > Given that those new results and clarifications will be integrated into the final version of the paper, I recommend acceptance, without further doubts on the soundness, and change my rating accordingly.

---

### Official Review · Reviewer_Autm · 2025-10-27

**Soundness:** 2
**Presentation:** 2
**Contribution:** 2
**Rating:** 4
**Confidence:** 3

**Summary:**

TurboQuant is proposed to minimize both mean-squared error and inner product distortion in vector quantization by using random rotation and optimal scalar quantizers, achieving near-optimal distortion rates across all bit-widths and dimensions, with a two-stage approach for unbiased inner product estimation, and experimental results validate its superiority over existing methods.

**Strengths:**

As detailed below, I suspect there may be issues with the  two major innovations regarding mean-squared error and inner product distortion.

**Weaknesses:**

The two major contributions claimed in the paper appear theoretically and logically untenable.

1) **$Q_{mse}$:** This paper does not impose restrictions on the distribution of vector x, nor does it specify whether the elements of x are independent. If the elements are not independent (commonly seen in natural data), then multiplying x by a random rotation matrix cannot guarantee mutual independence among the resulting vector's elements (contrary to the statement in Line 215), rendering simple scalar quantization methods (which quantize elements individually) inapplicable. Consequently, the paper's claimed optimal mse quantization would not hold.

2) **Unbiased $Q_{prod}$:** The multiplicative bias of $2/\pi$  (derived in Line 296) for the inner product $⟨x,y⟩$ is a constant and does not affect comparisons between different inner products. Therefore, I question the necessity of eliminating this bias by introducing complex quantization schemes for the residual vectors.

**Questions:**

Besides the two major concerns mentioned above, I have the following questions:

1) Line 210: Why is multiplication with a random rotation matrix employed? Is the purpose to achieve a beta distribution? Additionally, is this approach originally introduced in this paper?

2) The experiments are insufficient, as comparisons are made with only two quantization methods, and only a few bit-width cases are examined.

3) The “online” property claimed in the title is not explicitly demonstrated in either the theoretical analysis or experimental results.

---

> ### Author Response · Authors · 2025-11-23
> **Response to the review**
>
> We'd like to first thank the reviewer for their feedback. We address your questions/concerns below:
>
> `imposing restrictions on the distribution of vector x`:
> We wanted to clarify that in the proof of our distortion bounds we do not need independence of coordinates of x neither before nor after the random rotation and the statement in line 215 was just for providing an intuition about why the result is expected but the results proven in our Theorem 1, 2, 3 do not rely on independence of coordinates. We apologize if the statement is confusing and we will modify it to remove this confusion in the final version of the paper. In summary, our theorems prove that our quantizer achieves near optimal distortion bounds and there is no assumption made about the input vectors in order to prove these bounds.
>
> `multiplicative bias`:
> We agree the bias is not important in applications to ranking points by their scores. However, in other use-cases such as KV cache compression and model quantizations, this bias becomes critical as it can exponentially alter attention scores. We believe that the observation of optimizing for MSE loss introducing a dot-product bias is both intriguing and informative for readers, as is the design of residual vector quantization schemes. However we stay open to the reviewers suggestions.
>
> `reason for multiplication with a random rotation matrix`:
> Random rotation allows us to obtain optimal distortion rate for vectors while making no assumption about the vector distribution. Even if the vector has a worst case distribution with large outlier coordinates, after random rotation each coordinate will have a beta distribution which we know how to quantize optimally. The use of random rotation has been explored before (see references like Quarot, RabitQ, QJL, etc).
>
> `experiments are insufficient`:
> We have run a new, comprehensive set of experiments for both KV cache quantization and Approximate Nearest Neighbor (ANN) search, and we are committed to integrating these results into the final version of the paper.
> First we present the new KV cache quantization performance on the GSM8K mathematical reasoning dataset, using the Qwen2-7B-Instruct model (both model and dataset are new additions not present in the initial submission). This analysis also incorporates a comparison against a new baseline method, KV-Rotate, with results summarized in the table below:
>
> | Qwen2-7B-Instruct                  | GSM8K |
> |-------------------------|--------------:|
> | Exact (FP16) |85.7%       |
> | **TurboQuant (3-bits)**         | **84.3%**       |
> |KV-Rotate (3-bits) | 83.6%       |
> | KIVI (3-bits)       | 81.6%       |
>
> Next, we present new ANN experiments to benchmark our performance against more recent quantization papers, including NestQuant, LSQ++, and OPQ. For your initial review, we present the recall metrics on the GloVe ($d=200$) dataset with bit-width=4, and we commit to incorporating the results for the OpenAI datasets as well in the final version of the paper.
> | Method                  | Recall@1 | Recall@2 | Recall@4 | Recall@8 | Recall@16 | Recall@32 |
> |-------------------------|--------------:|-------------:|--------------:|------------------:|----------------:|----------------:|
> | **TurboQuant**         | **0.8829**       | **0.9756**      | **0.9978**       | **0.9999**           | 1.0         | 1.0         |
> |NestQuant | 0.8249       | 0.9460      | 0.9917       | 0.9988           | 1.0         | 1.0         |
> | LSQ++       | 0.8587       | 0.9664      | 0.9965       | 0.9993           | 1.0         | 1.0         |
> | OPQ       | 0.8229       | 0.9456      | 0.9956       | 0.9978           | 1.0         | 1.0         |
>
>
> `online property not explicitly demonstrated`:
> TurboQuant qualifies as an online quantization algorithm by virtue of its data-independent nature, which eliminates the need to learn or tune parameters based on a specific dataset. Consequently, in a streaming data environment, we can quantize each data point immediately upon its arrival. The KV cache is a prime example where this property is essential: keys and values are generated sequentially on the fly, demanding real-time quantization before being inserted into the cache. This immediate, on-the-spot processing is precisely the online property we attribute to TurboQuant. We apologize if this was not explicitly articulated in the initial submission and assure the reviewer that the final manuscript will be modified to provide greater clarity on this point.

---

> > ### Comment · Reviewer_Autm · 2025-11-24
> >
> > I would like to thank the authors for their  explanations and additional experiments. However, it appears that the authors have not fully addressed my  concerns on the two major contributions:
> >
> > 1) **Concerning the mutual independence of the random projection of $x$:** If the coordinates of the projected  $x$
> >  are not independent and the distribution of  $x$ is not preconditioned, how can the vector quantization problem be equivalently transformed or simplified into a scalar quantization problem? Is vector quantization equivalent to scalar quantization without any condition on data distributions?  This should be incorrect. Moreover, note that although the paper  title claims to investigate vector quantization, it have actually focused on scalar quantization.
> >
> > 2) **Regarding the multiplicative bias:** Given that the bias is a known constant value of $2/\pi$, why not simply subtract it directly? This approach would be much simpler than the method proposed in the paper.

---

> ### Author Response · Authors · 2025-11-26
> **Response to the confusion on coordinate independence**
>
> We thank the reviewer for probing the theoretical foundation of our quantization approach. We would like to first resolve the confusion on coordinate independence. Our response to the concerns about the multiplicative bias will follow in a separate comment.
>
> `mutual independence of the random projection of x`:
> The answer lies in the Linearity of Expectation and the Isometry of the random rotation, which allow us to minimize MSE by minimizing coordinate-wise distortion, without requiring the joint distribution to factorize (i.e., without requiring independence).
> 1. Orthogonal Invariance of MSE:
> Let $x$ be an arbitrary fixed input vector. We apply a random orthogonal matrix $\Pi$ to get $z = \Pi x$. We then scalar-quantize $z$ to get $\hat{z}$. The reconstruction is $\hat{x} = \Pi^\top\hat{z}$. Because orthogonal transformations preserve Euclidean norms, the distortion in the original space is identical to the distortion in the rotated space:
> $$\|x - \hat{x}\|^2 = \|\Pi^\top(z - \hat{z})\|^2 = \|z - \hat{z}\|^2 = \sum_{i=1}^d (z_i - \hat{z}_i)^2$$
> 2. Linearity of Expectation (The Key Mechanism):
> We seek to minimize the expected MSE over the randomness of $\Pi$. By linearity of expectation:
> $$\mathbb{E}[\|x - \hat{x}\|^2] = \sum_{i=1}^d \mathbb{E}[(z_i - \hat{z}_i)^2]$$
> To minimize this sum, it suffices to minimize each term $\mathbb{E}[(z_i - \hat{z}_i)^2]$ individually. This minimization depends only on the marginal distribution of each coordinate $z_i$. It does not require the joint distribution of $z_1, ..., z_d$ to be independent.
> 3. The Role of Random Rotation:
> While the input $x$ can have arbitrary distribution with weird dependencies, the random rotation $\Pi$ ensures that for any input $x$, the coordinates of $z = \Pi x$ are identically distributed (though not independent). Specifically, they follow a predictable Beta distribution (concentrated around 0) determined solely by the dimension $d$ and norm $\|x\|$ (see Lemma 1).
> Since the marginal distribution is known and fixed (Beta), we can design a single optimal scalar quantizer (Lloyd-Max) that minimizes $\mathbb{E}[(z_i - \hat{z}_i)^2]$.
> 4. Conclusion on "Equivalence":
> Therefore, the problem of minimizing the expected VQ distortion for $\mathbf{x}$ is mathematically equivalent to minimizing the SQ distortion for the Beta-distributed coordinates $z_i$. This holds strictly due to the geometry of the rotation, without assuming independence of the input $x$ or the projected vector $z$.
>
> **Regarding the Paper Title:**
> The method is a Vector Quantization scheme because it operates on the vector as a whole (applying a $d \times d$ rotation matrix) to exploit the geometry of the high-dimensional sphere. A pure Scalar Quantizer would process coordinates in isolation; our method couples them via rotation to "democratize" the energy and standard deviations, achieving vector-level distortion guarantees that pure scalar quantization on natural data cannot.
>
> **Key Remark:**
> The assertion in Line 215 regarding "near-independence" was primarily intended to provide an intuition on the efficiency of the quantization rate (an entropy argument); however, the mathematical argument for minimizing distortion (MSE) depends solely on the marginal moments, which is a significantly less stringent requirement. This initial intuition is, nevertheless, a strong approximation, since in high dimensions, the distribution of the rotated vector $z = \Pi x$ converges toward a Normal distribution. Because the covariance between any pair of coordinates in $z$ is zero, resulting in a covariance matrix for $z$ that is diagonal, the coordinates of $z$ are approximately independent (If z had a Normal distribution with diagonal covariance, coordinates of z would be independent). This intuitively explains why applying an optimal scalar quantizer to $z$ can achieve a nearly optimal distortion rate.
>
> We are committed to incorporating this detailed discussion into the final version of the paper to ensure maximal clarity for our readers.

---

> ### Author Response · Authors · 2025-11-27
> **Response to the concern about multiplicative bias**
>
> Regarding the multiplicative bias, we appreciate the reviewer’s suggestion to use simple rescaling (multiplying by $\pi/2$) to correct the bias. You are absolutely correct that if the goal is solely to obtain an unbiased estimator, simple rescaling is the standard and most efficient approach.
>
> However, we chose the residual quantization approach for a reason tied to the inner product error objectives of our work defined in Equation (2) of our paper. While rescaling by $\pi/2$ eliminates the bias, it strictly increases the variance (and consequently the inner product error) of the estimator quadratically. Therefore, fixing the bias via scaling amplifies the distortion by a factor $(\pi/2)^2$.
>
> On the other hand, the residual’s norm decays at an optimal rate as shown in Theorem 1, Thus applying an unbiased quantizer on the residual provides a fine-grained correction that further reduces the distortion, rather than amplifying the noise of the coarse estimate. This discussion can be made more formal and we assure you a formal discussion on this will be added to the final version of the paper in case of acceptance.

---

> ### Comment · Reviewer_Autm · 2025-11-28
>
> Thank you for your further explanations. I would like to **raise the score**, yet there seems to be no option to revise the score. I  hope the AC will take my opinions into account when formulating decisions.

---

### Official Review · Reviewer_WFrV · 2025-10-31

**Soundness:** 4
**Presentation:** 4
**Contribution:** 4
**Rating:** 10
**Confidence:** 5

**Summary:**

This paper proposes TurboQuant as method for low-distortion vector quantization. To minimize MSE, TurboQuant first randomly rotates the input vector and then uses k-means generated codebook for each dimension. For an unbiased estimation of inner product, TurboQuant combines the random rotation method for MSE with an unbiased 1-bit quantization method via a residual quantization approach. Theoretical results show that the quantization errors of TurboQuant are close to the optimization. Empirical results show that TurboQuant provides good performance for both KV cache compression and nearest neighbor search.

**Strengths:**

Thank you for the interesting paper! I think this paper is among the best vector papers I have read in this year, and I learned a lot from the paper.

S1: A comprehensive discussion for the applications of vector quantization is provided in the introduction.

S2: TurboQuant has a low quantization complexity and yet strong performance.

S3: The theoretical analysis is in-depth, and the theoretical results are strong.

S4: Combing different vector quantization methods via a residual quantization like approach, despite straightforward, is interesting and makes sense.

S5: The experiment results are strong and cover both KV cache compression and nearest neighbor search.

S6: The presentation is fluent. Although the paper is heavy on math, the author makes it easy to read even for readers that may not be familiar with this area.

**Weaknesses:**

The paper can be enhanced with more detailed discussions and experiments to compare TurboQuant with RaBitQ and variants (e.g., [1]). Note that I do not think RaBitQ affects the novelty of TurboQuant since the theorical results of TurboQuant are much stronger.

D1: RaBitQ is discussed in the appendix but the discussions are far from insufficient. The original RaBitQ quantization is slow due to the joint search for the optimal quantized bits over dimensions but a recent work [1] solves the problem. Moreover, [1] also uses PCA to significantly reduce the quantization error of RaBitQ. RaBitQ and variants are similar to TurboQuant in that they all use random projection; but they are also different in crucial aspects, (1) they search for quantized bits while TurboQuant uses k-means generated codebook independently for each dimension; (2) they use a re-normalization trick for unbiased inner product estimations while TurboQuant combines random rotation with another vector quantization method. A crucial question is that which design is better for the two purposes, and giving clear answers will be valuable for this area. Some ablation experiments can be added for these design choices, e.g., check the quantization error by using different design combinations. It will even better if some theorical analysis can be conducted.

[1] SAQ: Pushing the Limits of Vector Quantization through Code Adjustment and Dimension Segmentation

**Questions:**

NA

---

> ### Author Response · Authors · 2025-11-23
> **Response to the review**
>
> We thank you for your positive and constructive feedback and appreciate you bringing this new paper (SAQ) to our attention.
>
> Please note that while certain techniques in the referenced work, such as PCA, undeniably improve quality, they introduce data dependence. Our primary emphasis with TurboQuant remains data obliviousness, a property crucial for applications that need real time quantization like KV cache as well as seamless integration into distributed environments.
>
> Furthermore, these techniques are readily combinable with TurboQuant. For instance, PCA and dimension segmentation can be used to find subspaces of varying importance (determined by singular values), allowing for projecting the vectors into these subspaces and then applying TurboQuant with different bitwidths—assigning higher bitwidths to subspaces with larger singular values and lower ones to those with smaller singular values.
>
> Unfortunately, the code and implementation for this novel method are not yet publicly available. We plan to initiate contact with the authors of this paper soon and, upon receiving their code, we commit to running the requested experiments and incorporating them into the final version of our paper.

---

> > ### Comment · Reviewer_WFrV · 2025-11-26
> > **Comparing with the designs of RabitQ will make the paper more impactful**
> >
> > In my previous comment D1, I am not asking the authors to compare TurboQuant with SAQ, I am aware this is unfair because SAQ uses data distribution. I meant to compare the different design choices made by TurboQuant and RabitQ, i.e., (1) RabitQ searches for quantized bits while TurboQuant uses k-means generated codebook independently for each dimension; (2) RabitQ uses a re-normalization trick for unbiased inner product estimations while TurboQuant combines random rotation with another vector quantization method. The question is that whether the design choices of RabitQ or TurboQuant are better after applying random projection. I still think making these questions clear with benefit the field, and the code for RabitQ is easily accessible. Although this requires extra work, I strongly encourage the authors to add these experiments in the final version of the paper.

---

### Official Review · Reviewer_X81d · 2025-11-01

**Soundness:** 2
**Presentation:** 3
**Contribution:** 3
**Rating:** 6
**Confidence:** 3

**Summary:**

This paper introduces TurboQuant, an online vector quantization algorithm designed to speed up nearest-neighbor retrieval and vector search by continuously updating codebooks as new data arrives. The core idea is to maintain good quantization quality in a streaming setting by using lightweight updates rather than full re-training. The method provides theoretical guarantees on quantization error and memory usage and is evaluated on several ANN benchmarks to show improved trade-offs between accuracy, latency, and memory.

**Strengths:**

S1 Addresses a relevant and timely problem: efficient vector quantization in streaming/online settings, which is important for large-scale retrieval systems.

S2 Includes theoretical analysis that supports the stability and bounded error of the online updates.

S3 Empirical results show clear improvements over static or periodically-retrained quantization baselines in both speed and accuracy.

S4 Paper is generally well-written, and motivations for online updates vs. periodic retraining are clearly explained.

**Weaknesses:**

The paper is in general quite solid. My question is that the paper seems missing comparisons with stronger or more recent quantization methods, such as those using residual or product quantization variants with adaptive codebook updates, or learned quantizers from recent literature. It’s not entirely clear how TurboQuant performs relative to the current state of the art. Can you provide more literature review and comparisons?

**Questions:**

NA

---

> ### Author Response · Authors · 2025-11-23
> **Response to the review**
>
> We thank the reviewer for their constructive and insightful feedback.
>
> We'd like to start by elaborating on a feature of TurboQuant. One important feature is TurboQuant's "online" nature which stems from its use of a data-independent or data oblivious codebook that obviates the need for continuous updates as new data arrives. We achieve this by first normalizing the vectors by their l2 norm and then applying a random rotation to ensure the entries of the vectors will have a beta distribution post rotation. Then we show that there exists a universally optimal codebook for quantizing each coordinate with this beta distribution. Thus, there is no need to tune or update the codebook.
>
> `comparison with more recent methods`:
> First we'd like to mention that TurboQuant is a data oblivious method which is a very important property for distributed computing in applications such as distributed nearest neighbor search. The main purpose of our experiments was to show that being data independent does not really lead to worse quantization error and TurboQuant is even able to outperform data dependent product quantization methods and achieve higher recall for nearest neighbor search, especially against the FAISS baseline. We also wish to clarify that residual quantization, while relevant to partitioning centroids in ANN indices, is not fundamentally a quantization technique. Our method is fully compatible with residual vectors. In our experiments, we focused exclusively on the quantization problem, employing exact nearest-neighbor searches with quantized data points rather than an ANN index, thereby isolating the effects of quantization. Our quantization technique is readily applicable to residualized vectors within any index, including FAISS or SCANN.
>
> Furthermore, we ran new experiments to benchmark our performance against more recent quantization papers, including NestQuant, LSQ++, and OPQ. For your initial review, we present the recall metrics on the GloVe ($d=200$) dataset with bit-width=4, and we commit to incorporating the results for the OpenAI datasets as well in the final version of the paper.
> | Method                  | Recall@1 | Recall@2 | Recall@4 | Recall@8 | Recall@16 | Recall@32 |
> |-------------------------|--------------:|-------------:|--------------:|------------------:|----------------:|----------------:|
> | **TurboQuant**         | **0.8829**       | **0.9756**      | **0.9978**       | **0.9999**           | 1.0         | 1.0         |
> |NestQuant | 0.8249       | 0.9460      | 0.9917       | 0.9988           | 1.0         | 1.0         |
> | LSQ++       | 0.8587       | 0.9664      | 0.9965       | 0.9993           | 1.0         | 1.0         |
> | OPQ       | 0.8229       | 0.9456      | 0.9956       | 0.9978           | 1.0         | 1.0         |

---

### Public Comment · ~Jonas_Matthias_Kübler1 · 2026-03-26
**Reproducible Experiments**

Dear authors,
congratulations on your paper and the large public attention through your blogpost.

In the interest of science, it would be great to be able to verify your results and understand the experimental methodology better. However, with the code in the supplementary material I don't find enough instructions. Could you please update the supplement or provide a more comprehensive code? Any further information would be appreciated.

Also for the speedup experiment in Figure 2 c) could you please provide more detail on the methodology? Which head dimension $d$ does this consider? Is Q of shape $1xd$ and K of shape $nxd$? Also in the paper you say speedup is measured relative to PyTorch einsum baseline, whereas in the blog it is jax. Also in the blog you state the baseline is FP32. That seems not a good baseline, as I don't know of any framework that would not keep KV cache in max 16 bits by default.

Another question with respect to your LongBench and NIAH experiments. The way I understand your methodology, during prefill you use the Keys and Values in BF16. So there is no prefill speedup, and also no quality degradation at all during this prefill phase. Then the KV cache is quantized and used for the decoding. In this phase, there could be accuracy degradations, but also speedups. However, my understanding of the benchmarks you consider is that they are very prefill heavy. So overall I would expect that there is little accuracy degradation, but also no speedup.
Did you also consider benchmarks that have long generation phases? For example gpt-oss with reasoning high on GPQA or AIME25 could be really interesting. Another approach to investigate this could also be to run with chunked prefill, because then the corrupted KV cache actually get's used.

I also realized that in Table 1, the results for Llama TurboQuant with 2.5 bits got better wrt to your arxiv preprint (ICLR paper 49.74, arxiv 49.44). What did you change in between those version. you essentially closed the gap by 50%!




Thank you very much and see you at ICLR with more questions :)

Best,
Jonas

---

### Public Comment · ~Cheng_Long1 · 2026-03-27
**Concerns from the RaBitQ Authors Regarding Method Description, Theoretical Comparison, and Experimental Disclosure**

Dear ICLR community,

We the authors of the RaBitQ line of work [1, 2]. We are posting this comment to create a public record because the public discussion and promotion of TurboQuant have already created substantial confusion about its relationship to our RaBitQ line of work [1, 2]. These issues and explanations were not raised for the first time. In January 2025, Majid Daliri, the second author of the paper, contacted us to debug his Python translation of our RaBitQ implementation. In May 2025, after we came across their TurboQuant paper on arXiv, we raised the concerns below directly with him in detail. Despite that notice, the authors retained the inaccurate statements in their ICLR submission. Recently, on March 26, 2026, we formally notified all authors again. However, they agreed to fix only part of these issues and only after the ICLR 2026 conference takes place, which we believe is insufficient to dispel the widespread misunderstanding created by their recent promotion and may instead create further confusion at the ICLR meeting itself.

Our concern has three parts.

1. **Method-level description of RaBitQ is materially incomplete.** TurboQuant repeatedly describes random rotation as a key step of its method, yet its description of RaBitQ reduces mainly to a grid-based PQ framing while omitting the Johnson-Lindenstrauss transformation / random rotation, which is one of the most important linkage between the two methods. Moreover, even after two reviewers asked for clarification and discussion of the Johnson-Lindenstrauss transformation / random rotation, the ICLR camera-ready version of TurboQuant still did not add such a discussion; instead, the original description of RaBitQ in the main body was moved to the appendix.

2. **The theoretical description is not supported.** TurboQuant described RaBitQ's guarantees as "suboptimal" and attributed this to "loose analysis" without any explanations, although our paper [2] posted in September 2024 had already clearly claimed asymptotic optimality, which matches the optimal bound by Alon and Klartag [3]. Even after this issue was explicitly raised and clarified in emails in May 2025, the authors still do not provide a systematic explanation of how TurboQuant's guarantees compare to the RaBitQ line in their ICLR submission.

3. **The empirical comparison also lacks full disclosure.** Majid's January 2025 emails show that he had translated our C++ implementation of RaBitQ into Python and asked us to help debug it. In May 2025, he further acknowledged that, in the reported runtime setting, the RaBitQ baseline was run on a single CPU with multiprocessing disabled. The TurboQuant method itself is run on an A100 GPU. Yet the public paper makes efficiency claims without clearly disclosing that experimental setup. This issue was also raised in our private emails in May 2025.

In May 2025, our emails directly raised the theoretical and empirical issues; Majid wrote that he had informed his co-authors. During ICLR review, reviewers also asked for clarification about random rotation and the relation to RaBitQ. On March 26, 2026, we formally raised these concerns again to all authors and were told that corrections would wait until after the ICLR 2026 conference takes place; we were also told that they would not acknowledge the structural similarity regarding the Johnson-Lindenstrauss transformation. We do not consider that acceptable given the present level of public promotion and community confusion.

We are posting this comment so that the community has an accurate public record. We request that the authors publicly and promptly clarify the method-level relationship between TurboQuant and RaBitQ, the theory comparison, and the exact experimental conditions underlying the reported RaBitQ baseline. Given that these concerns were known before ICLR submission and before the current round of public promotion of TurboQuant, we believe it is necessary to bring these issues into the public discussion.

Regards,
Cheng
(on behalf of authors of RaBitQ papers)

References

[1] Jianyang Gao and Cheng Long, "RaBitQ: Quantizing High-Dimensional Vectors with a Theoretical Error Bound for Approximate Nearest Neighbor Search," Proceedings of the ACM International Conference on Management of Data (SIGMOD), 2024.

[2] Jianyang Gao, Yutong Gou, Yuexuan Xu, Yongyi Yang, Cheng Long, and Raymond Chi-Wing Wong, "Practical and Asymptotically Optimal Quantization of High-Dimensional Vectors in Euclidean Space for Approximate Nearest Neighbor Search," arXiv:2409.09913, Sep. 2024; later published in SIGMOD 2025.

[3] Noga Alon and Bo'az Klartag, "Optimal compression of approximate inner products and dimension reduction," 2017 IEEE 58th Annual Symposium on Foundations of Computer Science (FOCS), IEEE, 2017.

---

> ### Comment · Reviewer_WFrV · 2026-03-28
> **Reviewer Comment**
>
> As a reviewer of the TurboQuant paper, I gave a high rating to support this paper due to its theoretical analysis and experimental results. However, I also explicitly pointed out that both RaBitQ and TurboQuant both use random projection and requested the authors of TurboQuant to compare how the design differences between TurboQuant and RaBitQ affect performance. I believe the right academic practice is to thoroughly discuss the differences between RaBitQ and TurboQuant in the paper. However, checking the camera-ready version of TurboQuant, I am surprised to find that RaBitQ is mentioned only once in the experiment part of the main paper.
>
> Regards,
>
> Reviewer WFrV

---

> ### Public Comment · ~Wenqi_Guo3 · 2026-03-30
>
> This is a serious issue that deserves more attention. It's disheartening to see the people doing the actual foundational work get overlooked while large, influential organizations hype up their own results. And when people voice honest concerns, they get downvoted for going against the wave. At this point it feels less like science and more like a PR competition.

---

> ### Public Comment · ~Majid_Daliri1 · 2026-03-31
> **Technical Response: Clarifications Regarding TurboQuant**
>
> In response to recent commentary regarding our paper, "TurboQuant," we provide the following technical clarifications to correct the record.
>
> ---
>
> ## 1. Core Novelty vs. Standard Techniques
> TurboQuant did not derive its core method from RaBitQ. Random rotation is a standard, ubiquitous technique in quantization literature, pre-dating the online appearance of RaBitQ, e.g. in established works like https://arxiv.org/pdf/2307.13304, https://arxiv.org/pdf/2404.00456, or https://arxiv.org/pdf/2306.11987. The true novelty of TurboQuant lies in our derivation of the exact distribution followed by the coordinates of rotated vectors, which we use to achieve optimal coordinate-wise quantization.
>
> ## 2. Correction on RaBitQ Optimality
> While the optimality of RaBitQ can be deduced from its internal proofs, the paper’s main theorem implies that the distortion error bound scales as $4^{\Theta(-b)}$. Because a hidden constant factor within the exponent could scale the error exponentially, this formal statement did not explicitly guarantee the optimal bound. This led to our honest initial characterization of the method as suboptimal. However, after a careful investigation of their appendix, we found that a strict $4^{-b}$ bound can indeed be drawn. Having now verified that this optimality is supported by their deeper proofs, we are updating the TurboQuant manuscript to credit their bounds accurately.
>
> ## 3. Materiality of Experimental Benchmarks
> Runtime benchmarks are immaterial to our findings. TurboQuant’s primary contribution is focused on **compression-quality tradeoff**, not a specific speedup. The merit of our work rests on maintaining high model accuracy at extreme compression levels; even if the runtime comparison with RaBitQ was omitted entirely, the scientific impact and validity of the paper would remain mostly unchanged.
>
> ## 4. Observations on Timing
> TurboQuant has been publicly available on arXiv since April 2025, and one of its authors was in communication with RaBitQ authors even prior to that, as RaBitQ authors have acknowledged. Despite having nearly a year to raise these technical points through academic channels, these concerns were only raised after TurboQuant received widespread attention.
>
> ---
> ---
>
> ## *We are updating our arXiv version with our suggested changes implemented.*

---

> > ### Public Comment · ~Cheng_Long1 · 2026-04-02
> > **Research Integrity Concerns Regarding the TurboQuant Paper (1)**
> >
> > We respond to each of four points raised by the authors in turn.
> >
> > ----
> >
> > **1. On the description of RaBitQ and its relationship to TurboQuant**
> >
> > The authors' response does not directly respond to the concern we raised, which is about the accuracy of TurboQuant's description of RaBitQ itself. We must repeat our concerns in detail as follows.
> >
> > In January 2025, several months before the TurboQuant paper appeared on arXiv, Majid Daliri, proactively contacted us and asked for help debugging his own Python version translated from our RaBitQ C++ implementation. This indicates that the TurboQuant team has a clear understanding of the technical details of RaBitQ.  Yet, in the arXiv version they released in April 2025, and again in the version they submitted to ICLR 2026 in September 2025, they described RaBitQ as grid-based PQ while omitting the core random rotation step. An ICLR reviewer independently pointed this out in the review, writing: “RaBitQ and variants are similar to TurboQuant in that they all use random projection,” and explicitly requested a fuller discussion and comparison. Even so, in the camera-ready version of ICLR, the TurboQuant authors not only failed to add any real discussion of RaBitQ, but actually moved their already incomplete description of RaBitQ out of the main text and into the appendix.
> >
> > ----
> >
> > **2. On the correction of the "suboptimal" characterization**
> >
> > We appreciate the authors' acknowledgment that RaBitQ's error bound is optimal. However, we must point out that we have raised the issues and clarified it to the TurboQuant team in May 2025, which is several months before the submission deadline of ICLR 2026.
> >
> > Our paper (arXiv:2409.09913, September 2024) explicitly claimed asymptotic optimality matching the Alon-Klartag bound in its abstract and stated contributions. We further raised this specific issue in detail in our emails to Majid Daliri in May 2025, providing a full technical clarification. Majid Daliri confirmed in writing that he had informed all co-authors. Despite this, the characterization of RaBitQ as "suboptimal" was retained without correction in the ICLR submission, throughout the review process, and in the camera-ready version.
> >
> > ----
> >
> > **3. On the experimental comparison and its disclosure**
> >
> > The authors' response does not directly respond to the concern we raised, which is about the deliberately created unfair experimental setup. We must repeat our concerns in detail as follows.
> >
> > Majid's January 2025 emails show that he had translated our C++ implementation of RaBitQ into Python. In May 2025, he further acknowledged that, in the reported runtime setting, the RaBitQ baseline was run on a single-core CPU with multiprocessing disabled. The TurboQuant method itself is run on an A100 GPU. Yet the public paper makes efficiency claims without clearly disclosing that experimental setup. This issue was also raised in our private emails in May 2025.
> >
> > Moreover, Google's recent promotion of TurboQuant has specifically highlighted the speed-up of the method, for example, “Introducing TurboQuant: Our new compression algorithm that reduces LLM key-value cache memory by at least 6x and delivers up to 8x speedup, all with zero accuracy loss, redefining AI efficiency” [4]. This indicates that efficiency is a core target of the TurboQuant project. This is contradictory to the authors’ response.
> >
> > [4] Google Research’s post on Linkedin: https://www.linkedin.com/feed/update/urn:li:share:7442298961455067136/?origin
> >
> > ----
> >
> > **4. On the timing and history of our concerns**
> >
> > The authors' claim that "these concerns were only raised after TurboQuant received widespread attention" is factually incorrect and requires direct correction.
> >
> > The timeline of our actions is as follows.
> > - In May 2025, we raised our concerns in detail directly with Majid Daliri by email. Majid engaged with these points over multiple exchanges and confirmed in writing that he had informed his co-authors in May 2025.
> > - In November 2025, after seeing that the ICLR submission retained the same factual issues, we wrote to the ICLR Programme Chairs to raise our concerns formally.
> > - In March 2026, after seeing both the wide-scale public promotion of TurboQuant and the camera-ready version — which still retained the same issues — we formally notified all authors of TurboQuant again in writing, contacted the ICLR chairs again, and subsequently posted this public comment.
> >
> > At every stage, we raised our concerns through the appropriate private or institutional channels first. We contacted the authors directly, then the venue chairs, then the authors again. We made this comment public only after all of these steps had failed to produce any correction across three successive versions of the paper — the arXiv version, the ICLR submission, and the camera-ready. The suggestion that we delayed raising concerns for strategic reasons inverts the documented sequence of events entirely.

---

> > > ### Public Comment · ~Cheng_Long1 · 2026-04-02
> > > **Research Integrity Concerns Regarding the TurboQuant Paper (2)**
> > >
> > > We are disappointed to see that the TurboQuant team has not directly responded to our concerns majorly. Their reply even suggests that we had not raised these technical points to them through academic channels over the past year, which is factually incorrect.
> > >
> > > We have submitted our email records with the TurboQuant team to ICLR Chairs. According to ICLR Code of Ethics “Researchers must not deliberately make false or misleading claims, fabricate or falsify data, or misrepresent results. Methods and results should be presented in a way that is transparent and reproducible. ”, we respectfully request that ICLR initiates a formal research-integrity review of this paper.

---

> > ### Public Comment · ~Ran_Ben-Basat1 · 2026-04-22
> > **With regard to point (1)**
> >
> > We want to highlight that the core novelty you highlight (analysis of the coordinate distribution after rotation) appears in the  DRIVE  paper (Lemma 8, Appendix A.4 of the supplemental material).
> > Moreover, as we acknowledge in DRIVE, this and more general results about the post-rotation distribution were already known (e.g., see eq. 3.3 in  Paweł J Szabłowski. [Uniform Distributions on Spheres in Finite Dimensional Lα and Their Generalizations. Journal of multivariate analysis](https://www.sciencedirect.com/science/article/pii/S0047259X97917188), 64(2):103–117, 1998).

---

> ### Public Comment · ~Wenqi_Guo3 · 2026-04-03
>
> > Runtime benchmarks are immaterial to our findings. TurboQuant’s primary contribution is focused on compression-quality tradeoff, not a specific speedup. The merit of our work rests on maintaining high model accuracy at extreme compression levels; even if the runtime comparison with RaBitQ was omitted entirely, the scientific impact and validity of the paper would remain mostly unchanged.
>
> If you show the number in the paper (which is in the appendix), you need to show your benchmark env. If you used a single core CPU for the baseline and a GPU for your method, you have to report it. Even if that is not your main contribution, you showed the number without context, which is misleading at least and data fabrication at worst.
>
> Also, it is not just an unfair comparison by limitations of GPU implementations; authors translated the C++ RaBitQ into Python. This means that authors already chose to re-implement it, and yet they did choose to implement it in a slow language (Python) with no GPU supports. Even if CPU is the only option, a single core with disabled MP support is a choice authors need to make out of the way as it's barely standard benchmark env for CPU runtime.
>
> The paper also says RaBitQ does not support vectorization, which I am not sure, but seems like RaBitQ has a SIMD-based version.

---

### Public Comment · ~Ran_Ben-Basat1 · 2026-04-22
**A Note on TurboQuant and the Earlier DRIVE/EDEN Line of Work**

To the ICLR Community,

We believe the current paper materially understates its relationship to earlier works [DRIVE (NeurIPS 2021)](https://proceedings.neurips.cc/paper/2021/hash/0397758f8990c1b41b81b43ac389ab9f-Abstract.html) and [EDEN (ICML 2022)](https://proceedings.mlr.press/v162/vargaftik22a.html).

The framework used by TurboQuant, of randomly rotating the input vector, mapping it to the closest codeword from an offline-computed codebook (by running the Lloyd-Max algorithm on the shifted-beta/normal distribution), and applying the inverse rotation by the dequantizer, was introduced and analyzed by the works above.
The current OpenReview version does not cite DRIVE or EDEN.
We think the overlap in ideas and analysis should be acknowledged.

In fact, as detailed in our [recent note](https://arxiv.org/pdf/2604.18555), using EDEN instead of TurboQuant performs better empirically on the TurboQuant paper's tasks/experiments, and has stronger theoretical guarantees, and there are clear and natural reasons for that.
TurboQuant_mse is a special case of DRIVE and EDEN obtained by fixing DRIVE/EDEN's scale parameter to S=1, and TurboQuant_prod can be viewed as a biased (b−1)-bit DRIVE/EDEN step (with the suboptimal S=1) followed by a 1-bit QJL quantization of the residual. As we also show, both theoretically and empirically, this construction is generally suboptimal relative to DRIVE/EDEN, often by more than a bit, e.g.,  1-bit, 2-bit, and 3-bit unbiased EDEN are more accurate than 2-bit, 3-bit and 4-bit TurboQuant-prod respectively.

We reached out privately to the authors in March 2026 to discuss these issues but did not receive a response, so we are posting this publicly in the interest of accurate attribution and clear comparison to prior art. For further details and reproduced experiments, see our note: [https://arxiv.org/pdf/2604.18555](https://arxiv.org/pdf/2604.18555).

---

### Public Comment · ~Jianyang_Gao1 · 2026-04-22
**Follow-up: arXiv technical report and reproducibility concerns**

Dear ICLR community and TurboQuant authors,

We are posting this follow-up because ICLR 2026 begins on April 23, 2026, and the public record surrounding TurboQuant still has not been visibly corrected.

We have released a technical report and reproduction code:

- Technical report: **Revisiting RaBitQ and TurboQuant: A Symmetric Comparison of Methods, Theory, and Experiments**
  https://arxiv.org/abs/2604.19528
- Reproduction code:
  https://github.com/VectorDB-NTU/rabitq-turboquant-comparison

The report compares RaBitQ and TurboQuant under a common framework, covering methodology, theoretical guarantees, and experiments. The experimental results reinforce and substantially extend our earlier concerns. They show that the TurboQuant paper materially understates RaBitQ's performance, and that the claimed empirical advantage of TurboQuant over RaBitQ cannot be reproduced from the released artifacts.

In directly comparable inner-product estimation and nearest-neighbor search settings, TurboQuant performs worse than RaBitQ in most tested configurations. In nearest-neighbor search, RaBitQ consistently achieves higher recall than both TurboQuant variants across the tested datasets and bit widths. For quantization time, when RaBitQ is evaluated on multi-core CPUs with normal multi-threading enabled, it is several orders of magnitude faster than the RaBitQ runtime reported in the TurboQuant paper. Conversely, when TurboQuant is evaluated using the released implementation on the stated A100 hardware, we observe quantization times up to approximately two orders of magnitude slower than those reported in the TurboQuant paper. These results indicate that the comparison in the TurboQuant paper materially underreports RaBitQ and gives readers an inaccurate impression of TurboQuant's advantage.

We emphasize again that we expressed our concerns with the TurboQuant authors in May 2025, months before the ICLR submission. Nevertheless, even as the ICLR conference begins, there is still no visible correction in the OpenReview record or the arXiv record.

To avoid further confusion during ICLR, we respectfully ask the TurboQuant authors to present the disputed RaBitQ comparison claims with clear caveats, rather than as settled conclusions, until the reproducibility issues are resolved. We would also appreciate visible clarifications in relevant public channels, including OpenReview, the arXiv/manuscript record, ICLR poster or presentation materials, and public-facing blog or social-media materials, so that readers can understand which claims remain disputed and what experimental conditions underlie them. To help the community resolve these discrepancies constructively, we encourage the TurboQuant authors to provide complete scripts, configurations, hardware details, and random seeds sufficient for reproducing their reported results.

Regards,
Jianyang Gao

---

### Meta-Review · Area_Chair_rDFk · 2026-01-04

**Summary:**

The reviewers had two main concerns regarding the validity of some theoretical claims for this work and comparisons with additional/stronger baselines. Both claims have been addressed, and the corresponding reviewers requested to increase their scores. Overall the paper seems solid and I recommend acceptance as poster

**Reviewer Concerns:**

All major concerns were addressed, including comparisons with stronger baselines and clarification of the theoretical validity of the approach.

**Reviewer Scores:**

Two reviewers with lower scores (4) either recommended acceptance or asked to increase their scores after rebutall but it was not possible. Hence, I believe all reviewers would have been positive for this work and wouldn't object acceptance as poster in the conference

---

### Decision · Program_Chairs · 2026-01-26

Accept (Poster)